# Rhythmic glucose metabolism regulates the redox circadian clockwork in human red blood cells

Ratnasekhar Ch [1,7,11], Guillaume Rey [1,8,11], Sandipan Ray[2,3,9], Pawan K. Jha[2,3], Paul C. Driscoll [4], Mariana Silva Dos Santos [4], Dania M. Malik[2,3], Radoslaw Lach[1,10], Aalim M. Weljie[2,3], James I. MacRae [4], Utham K. Valekunja [2,3] & Akhilesh B. Reddy[2,3,5,6 ✉]

Circadian clocks coordinate mammalian behavior and physiology enabling organisms to anticipate 24-hour cycles. Transcription-translation feedback loops are thought to drive these clocks in most of mammalian cells. However, red blood cells (RBCs), which do not contain a nucleus, and cannot perform transcription or translation, nonetheless exhibit circadian redox rhythms. Here we show human RBCs display circadian regulation of glucose metabolism, which is required to sustain daily redox oscillations. We found daily rhythms of metabolite levels and flux through glycolysis and the pentose phosphate pathway (PPP). We show that inhibition of critical enzymes in either pathway abolished 24-hour rhythms in metabolic flux and redox oscillations, and determined that metabolic oscillations are necessary for redox rhythmicity. Furthermore, metabolic flux rhythms also occur in nucleated cells, and persist when the core transcriptional circadian clockwork is absent in *Bmal1* knockouts. Thus, we propose that rhythmic glucose metabolism is an integral process in circadian rhythms.

[1] University of Cambridge Metabolic Research Laboratories, Institute of Metabolic Science, University of Cambridge, Addenbrooke's Hospital, Cambridge CB2 0QQ, UK. [2] Department of Systems Pharmacology and Translational Therapeutics, Perelman School of Medicine, University of Pennsylvania, Pennsylvania, PA 19104, USA. [3] Institute for Translational Medicine and Therapeutics, Perelman School of Medicine, University of Pennsylvania, Philadelphia, PA 19104, USA. [4] The Francis Crick Institute, 1 Midland Road, London NW1 1AT, UK. [5] Institute for Diabetes, Obesity, and Metabolism, University of Pennsylvania Perelman School of Medicine, Philadelphia, PA 19104, USA. [6] Chronobiology and Sleep institute (CSI), Perelman School of Medicine, University of Pennsylvania, Philadelphia, PA 19104, USA. [7] Present address: School of Biological Sciences, Queen's University Belfast, Belfast BT9 5DL, UK. [8] Present address: Unilabs Genetics Laboratory, 1003 Lausanne, Switzerland. [9] Present address: Department of Biotechnology, Indian Institute of Technology Hyderabad, Kandi, Sangareddy 502285, Telangana, India. [10] Present address: Department of Oncology, University of Cambridge, Cambridge Biomedical Campus, Cambridge CB2 0XZ, UK. [11] These authors contributed equally: Ratnasekhar Ch, Guillaume Rey. ✉email: areddy@cantab.net

Circadian rhythms allow species to adapt their physiology to daily environmental variation tied to the Earth's rotation[1]. They govern essential processes, including the regulation of hormones, the sleep-wake cycle, cell division, and immunity[2,3]. Disruption of circadian rhythms is associated with metabolic syndrome, obesity, type 2 diabetes and various neurological disorders[4,5]. Circadian rhythms are thought to be driven by transcriptional-translational feedback loops (TTFLs), whereby rhythmic expression of clock gene products regulate the expression of associated genes in an approximately 24-h cycle[6]. However, genetic studies of circadian clocks in various model organisms (cyanobacteria, *Drosophila*, *Arabidopsis*, mice, and humans) has shown that most clock genes are not evolutionarily conserved across distinct phylogenetic kingdoms, and TTFL components are not shared between organisms[7]. Several studies implicate non-transcriptional components in circadian rhythmicity. These include post-translational circadian oscillation of cyanobacterial KaiC phosphorylation and cytosolic mechanisms such as $Ca^{2+}$/cAMP[8,9]. However, there is limited homology of these posttranslational clock components across various model organisms[10], and thus it has been challenging to study posttranslational aspects of circadian biology in nucleated mammalian cells.

We previously found circadian rhythms in reduction-oxidation (redox) cycles of peroxiredoxin proteins in mammalian red blood cells (RBCs), which do not have nuclei and do not produce new RNA or protein[11]. We also found similar posttranslational peroxiredoxin rhythms in various prokaryotes and eukaryotes commonly used as circadian model systems[10,12]. This suggests that redox processes are important in circadian rhythm generation across phylogenetic boundaries, and may have evolved ~2.5 billion years ago with specific mechanisms to counteract the deleterious consequences of oxidative stress[7,10]. However, what drives circadian timing of redox balance is unclear. Metabolism is one such important player that may be a connecting node between redox processes and the clockwork.

In this study, we sought to understand the metabolic basis of peroxiredoxin oscillations in human red blood cells. To do this, we characterized circadian regulation of metabolites in human red blood cells using metabolomics and computed metabolic fluxes. Importantly, we uncovered rhythmic oscillations in glycolysis and pentose phosphate pathway metabolites and metabolic fluxes with distinct and opposite phases. Moreover, we find that circadian regulation of metabolic switching in glycolysis and pentose phosphate pathways is tightly coupled to redox oscillations. Perturbing key components of these metabolic pathways results in loss of redox rhythms in human RBCs. We then examined nucleated mammalian cells and found analogous metabolic oscillations, with and without the core TTFL clock feedback circuit. Thus, we find that circadian metabolic flux rhythms persist regardless of whether the TTFL is present.

## Results

**Metabolomics reveals circadian regulation of metabolites in human RBCs.** Because human RBCs cannot perform transcription and/or protein synthesis, they completely rely on metabolic processes to combat reactive oxygen species (ROS), created by auto-oxidation of hemoglobin, for survival[11,13,14]. As such, we hypothesized that metabolism exhibits non-transcriptional oscillations in RBCs. To determine whether metabolism exhibits a daily rhythm, we first analyzed the RBC metabolome in samples from human subjects. We collected fresh blood samples and incubated purified RBCs in constant conditions (at 37 °C in continuous darkness) ex vivo to exclude any temporal cues that could drive oscillations (Fig. 1A)[11,15–17]. Thus, any measurements

made reflect self-sustained rhythms in the isolated cells. We sampled cells every 3 h over two days and performed untargeted metabolite profiling by gas chromatography (GC) and liquid chromatography (LC) mass spectrometry (MS). We detected 1,698 features that had a coefficient of variation <30% in quality control samples. This included 533 features from negative mode LC-MS and 1074 from positive mode, with an additional 91 from GC-MS (Supplementary Fig. 1). We identified 43 rhythmic metabolites ($p_{adj} < 0.05$) using Rhythmicity Analysis Incorporating Nonparametric methods (RAIN)[18] (Fig. 1B, and Supplementary Table 1).

The majority of rhythmic metabolites included carbohydrates and redox metabolites. Overall, 20% (43 of 213 known detected metabolites) of the RBC metabolome displayed a 24-h cycle. This is consistent with the circadian metabolome of human blood plasma (~15–22%)[19,20]. Since RBCs cannot perform fatty acid synthesis, protein synthesis, the citric acid cycle, or nucleotide synthesis, these aspects of metabolism are not under circadian control. Principal Components Analysis (PCA) revealed that the corresponding time points from the first and second days grouped together, and that the major component separating the cycling metabolome was time of the day (Fig. 1C). Pathway enrichment analysis of rhythmic metabolites identified an enrichment in glycolysis and PPP pathways (false discovery rate, FDR < 0.01), but not others (Fig. 1D, E).

Glucose displayed a well-defined 24-h oscillation ($P = 0.005$ by RAIN), with glucose-6-phosphate (G6P) showing a similar phase, albeit with reduced rhythmicity ($P = 0.0369$ by RAIN) (Fig. 2A). Importantly, ribose-5-phosphate (R5P), a principal metabolite in the PPP, exhibited daily cycling in anti-phase to G6P (Fig. 2A, B). In addition, many other metabolites in the glycolytic pathway showed robust circadian cycles over at least two days (Fig. 2A, Supplementary Table 1). Together, these data show that there is circadian oscillation of specific glycolytic and PPP metabolites in human RBCs.

**Rhythmic glycolysis and PPP fluxes in human RBCs.** Metabolites levels are dependent on the input of fluxes through metabolic pathways, and these metabolic pathway fluxes are considered a crucial biological function, and are under strong evolutionary selection procedure[21,22]. To determine how rhythmic abundance of metabolites occurs, and the enrichment of metabolites in different circadian phases, we assessed the rate of flow through glycolysis and PPP. We employed an established labeling model to measure metabolic fluxes in human RBCs by nuclear magnetic resonance spectroscopy (NMR)[23]. Under constant conditions, as described above, we took samples at 4 h intervals over three days to determine flux through glycolysis and PPP (Fig. 3A). We collected samples at least 24-h after incubating cells in 11 mM labeled glucose. When 2-$^{13}C_1$-glucose is metabolized by glycolysis, it produces lactate labeled at position 2 (2-$^{13}C_1$-lactate; C2-lactate). In contrast, 3-$^{13}C_1$-lactate (C3-lactate) is produced when the labeled glucose passes through PPP (Fig. 3B). We quantified these differentially labeled lactates at a chemical shift ($\delta$) of 71.2 (C2-lactate) and 22.8 (C3-lactate) using NMR (Fig. 3B). We then calculated fluxes through PPP and glycolysis. We found circadian oscillation of glycolysis ($P = 0.009$ by RAIN), and PPP in anti-phase, for three consecutive days (Fig. 3C).

Peroxiredoxin oxidation (PRDX-SO$_{2/3}$) in RBCs from the same samples demonstrated circadian rhythms (Fig. 3D, Supplementary Figs. 2 and 3). We validated circadian flux measurements by performing an independent experiment with 11 mM 1,2-$^{13}C_2$-Glucose, assayed by GC-MS (Fig. 3E–G). When 1,2-13$C_2$-glucose is administered to RBCs, glycolysis generates m + 2 lactate, while PPP generates m + 1 lactate (Fig. 3E)[24]. GC-MS also

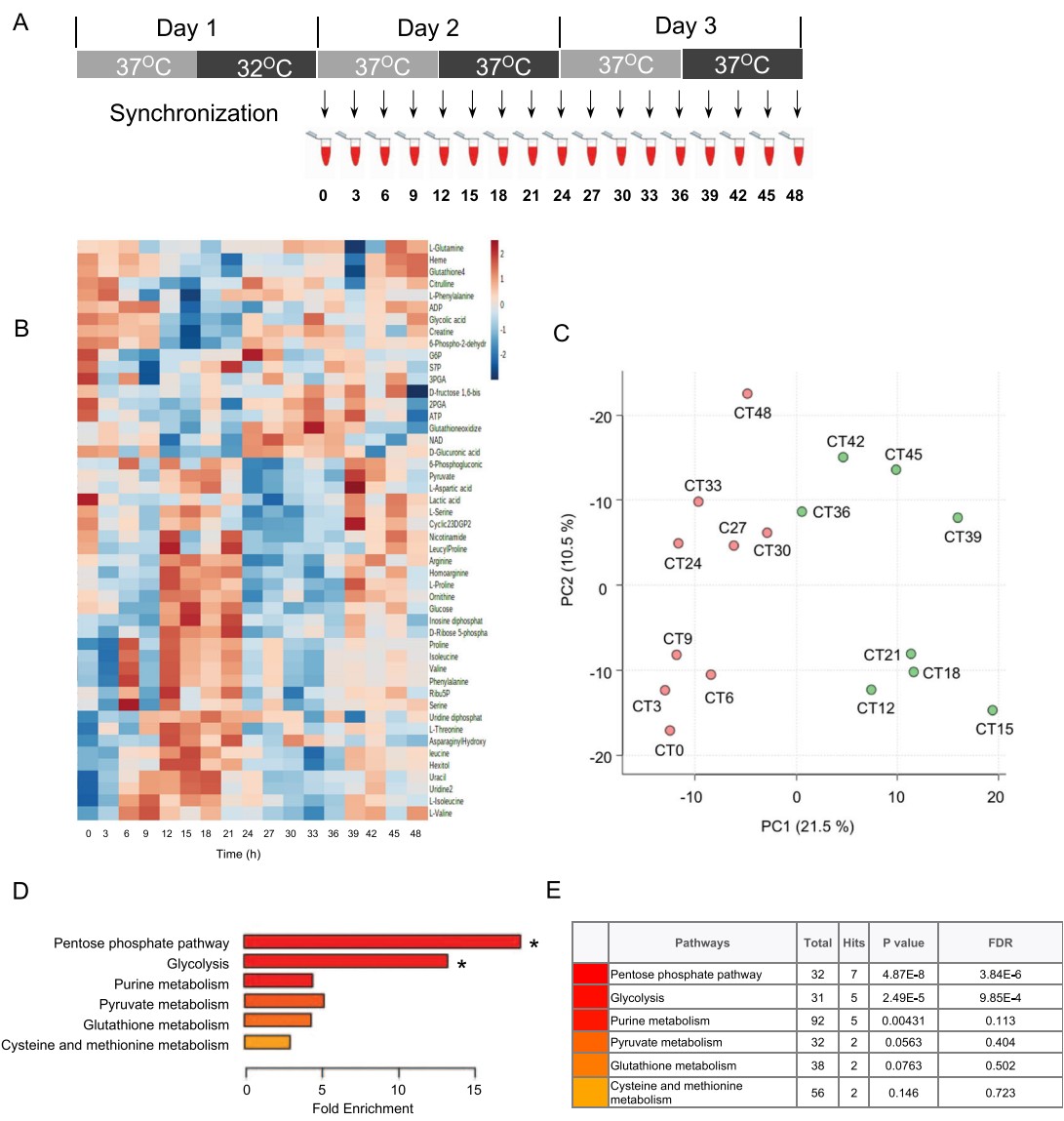

**Fig. 1 Redox and metabolic rhythms in human red blood cells (RBCs). A** Experimental scheme for untargeted Human RBCs circadian sampling. RBCs from $n = 3$–4 human subjects were incubated with 11 mM glucose and kept under constant conditions (37 °C in continuous darkness) and sampling was performed every 3 h. Untargeted metabolite analysis was performed using GC-MS and LC-MS. **B** Heatmap of all cycling features ordered by the phase of the oscillation. Features extracted from spectra using Progenesis QI (LC-MS) and MassHunter (GC-MS) were used for the rhythmicity analysis using RAIN algorithm (*$P < 0.05$, FDR = 0.2). Data are Z-score normalized log10 intensities. **C** Principal components analysis (PCA) of the cycling metabolome. The two major components separate the data into samples corresponding to the circadian day (left, orange) and night (right, green) time-points. **D** Pathway analysis performed using Metabolite Set Enrichment Analysis (MSEA) using only identified 24-h rhythmic metabolites. The color bar represents the significance level, as shown in the table. *FDR < 0.01. **E** Table representing the enriched metabolic pathways with the corresponding metabolite hits, P values and FDR.

demonstrated circadian fluxes through glycolysis and PPP ($P = 0.042$ by RAIN; Fig. 3F). Corresponding peroxiredoxin oxidation (PRDX-SO$_{2/3}$) immunoblots also revealed circadian rhythms (Fig. 3G and Supplementary Figs. 4 and 5). Indeed, NMR and GC-MS readouts are very similar when compared side-by-side (Fig. 3H).

PPP flux peaked during the circadian day (phase ~11) whereas glycolytic flux peaked at night (phase ~23). Peroxiredoxin oxidation was highest during the day, aligned with the PPP flux peak (phase ~13). We saw a progressive decrease in glucose in the surrounding medium in our single pulse labeling experiments (Supplementary Fig. 6). Because this could impact measurements in flux models, we next performed a dynamic glucose pulse experiment using two different tracers: 11 mM 1,2-$^{13}$C$_2$-glucose

or 1-$^{13}$C$_1$-glucose (Fig. 4A, B). In concordance with our results using a single pulse of glucose at the beginning of the experiment, we found circadian oscillations of glycolytic flux using the pulsed glucose time courses (Fig. 4A, B).

Circadian rhythms in mouse RBCs have also been reported[16],[17]. To determine if metabolic flux varies over the day in circulating RBCs in vivo, we took blood from mice at opposite phases of the circadian cycle—in the middle of the subjective day (CT06) or night (CT18)—and then labeled RBCs with glucose and determined flux by GC-MS (Fig. 4D). As in our ex vivo experiments, we found at least a 2-fold difference in flux in labeled cells from opposite circadian phases (Fig. 4D). Collectively, these results indicate that there is a circadian rhythm of metabolic flux through glycolysis and PPP in red cells, both

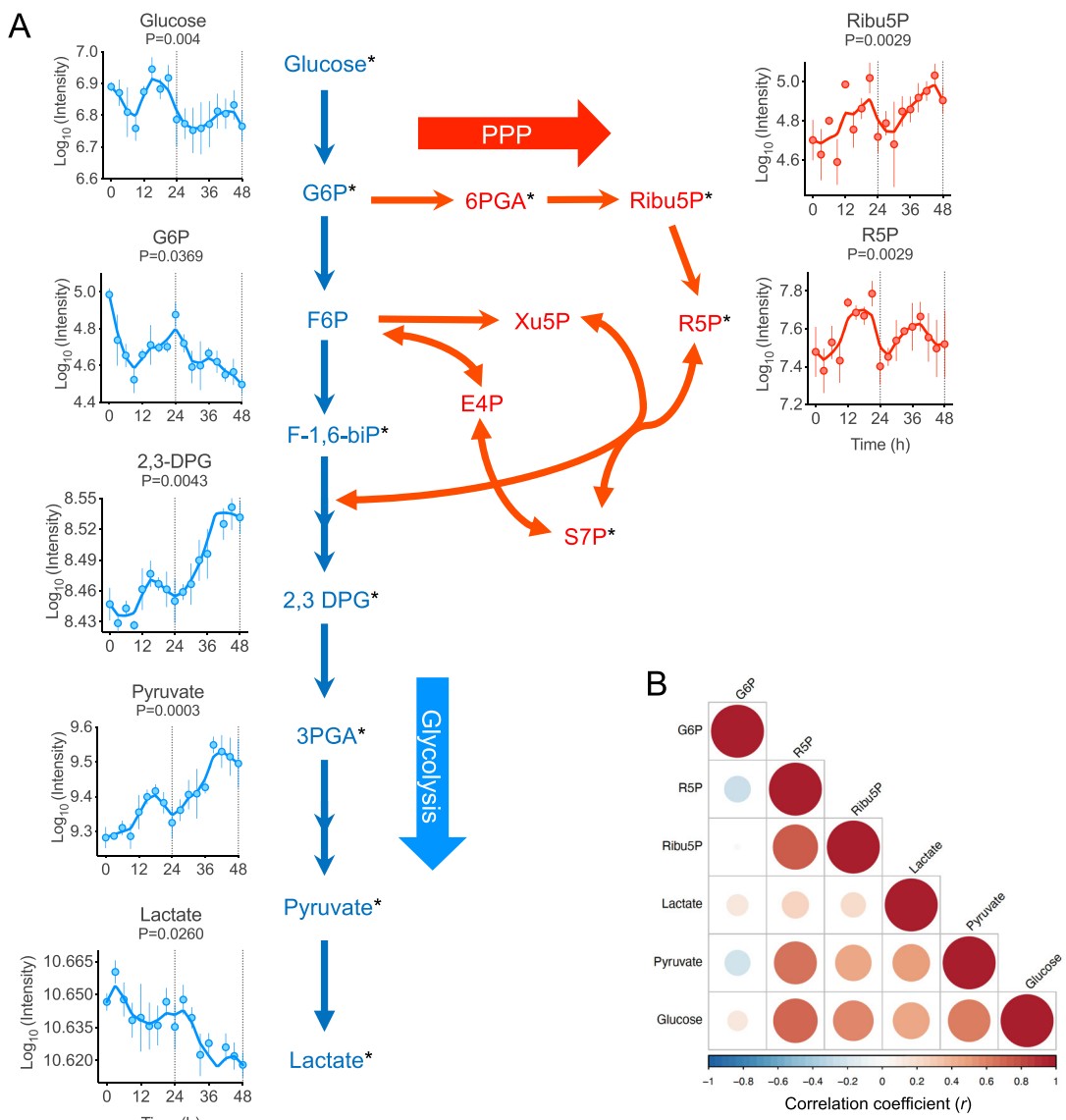

**Fig. 2 Rhythmic accumulation of central carbon metabolites in human red blood cells (RBCs). A** Selected significant rhythmic metabolites (*$P < 0.05$) in human RBCs analyzed by an untargeted metabolomics approach using GC-MS and LC-MS platforms. Rhythmicity analysis was performed using RAIN algorithm (*$P < 0.05$, FDR = 0.2). RBCs were synchronized for 24 h and then sampling was performed for every 3 h over a period of 48 h. Relative log10 normalized intensities for metabolites are presented. Glycolytic metabolites are represented in blue and PPP metabolites in red. List of identified rhythmic metabolites presented in Supplementary Table 1. Glucose ($n = 3$ replicates), G6P = glucose-6-phosphate ($n = 4$ replicates), 2,3-DPG = 2,3-diphosphoglycerate ($n = 3$ replicates), Ribu5P = ribulose-5-phosphate ($n = 4$ replicates), R5P = ribose-5-phosphate ($n = 3$ replicates), pyruvate ($n = 3$ replicates), lactate ($n = 3$ replicates). **B** Correlation plot comparing selected glycolytic and PPP metabolites. Data are mean ± s.e.m.

in vivo and ex vivo, which may drive circadian metabolite profiles that we uncovered in these pathways.

**Redox couples cellular metabolism in human RBCs**. Peroxiredoxins in red blood cells play a crucial role to maintain redox balance by neutralizing $H_2O_2$ generated in part by auto-oxidation of hemoglobin[14]. Peroxiredoxin oxidation exhibits circadian rhythms, prompting the question of how coupling between metabolic cycles and redox state is achieved. To understand this link, we investigated the interrelationship between glucose-metabolizing pathways in RBCs and peroxiredoxin redox state. We first inhibited flux through glycolysis using heptelidic (koningic) acid (HA), a potent inhibitor of glyceraldehyde 3-phosphate dehydrogenase (GAPDH)[25,26] (Fig. 5A). Metabolic inhibition experiments were performed on RBCs isolated from fresh blood samples collected from human subjects and metabolic

inhibitors were added at the start of the experiment (Fig. 5B). We found that fluxes through glycolysis and PPP became arrhythmic ($P = 0.865$ by RAIN) after treatment with 1.9 μM HA (Fig. 5C; only glycolytic flux shown for clarity).

We then inhibited flux through PPP using 10 mM 6-aminonicotinamide (6AN), an inhibitor of the two NADPH-producing enzymes in PPP (Fig. 5A)[27,28]. The amplitude of flux oscillations was reduced and non-rhythmic ($P = 0.127$ by RAIN; Fig. 5C). Peroxiredoxin oxidation in cells treated with these inhibitors was also affected, abolishing 24-h patterning (Fig. 5D, Supplementary Fig. 7). Interestingly, although HA treatment caused arrhythmicity ($P = 0.231$ by RAIN), 6AN prolonged period to 36 h ($P < 0.001$ by RAIN) compared to control cells (Fig. 5D). Importantly, the inhibitors did not affect RBC viability at the doses used (Supplementary Fig. 8), and they caused differential changes in the RBC metabolome, demonstrating their

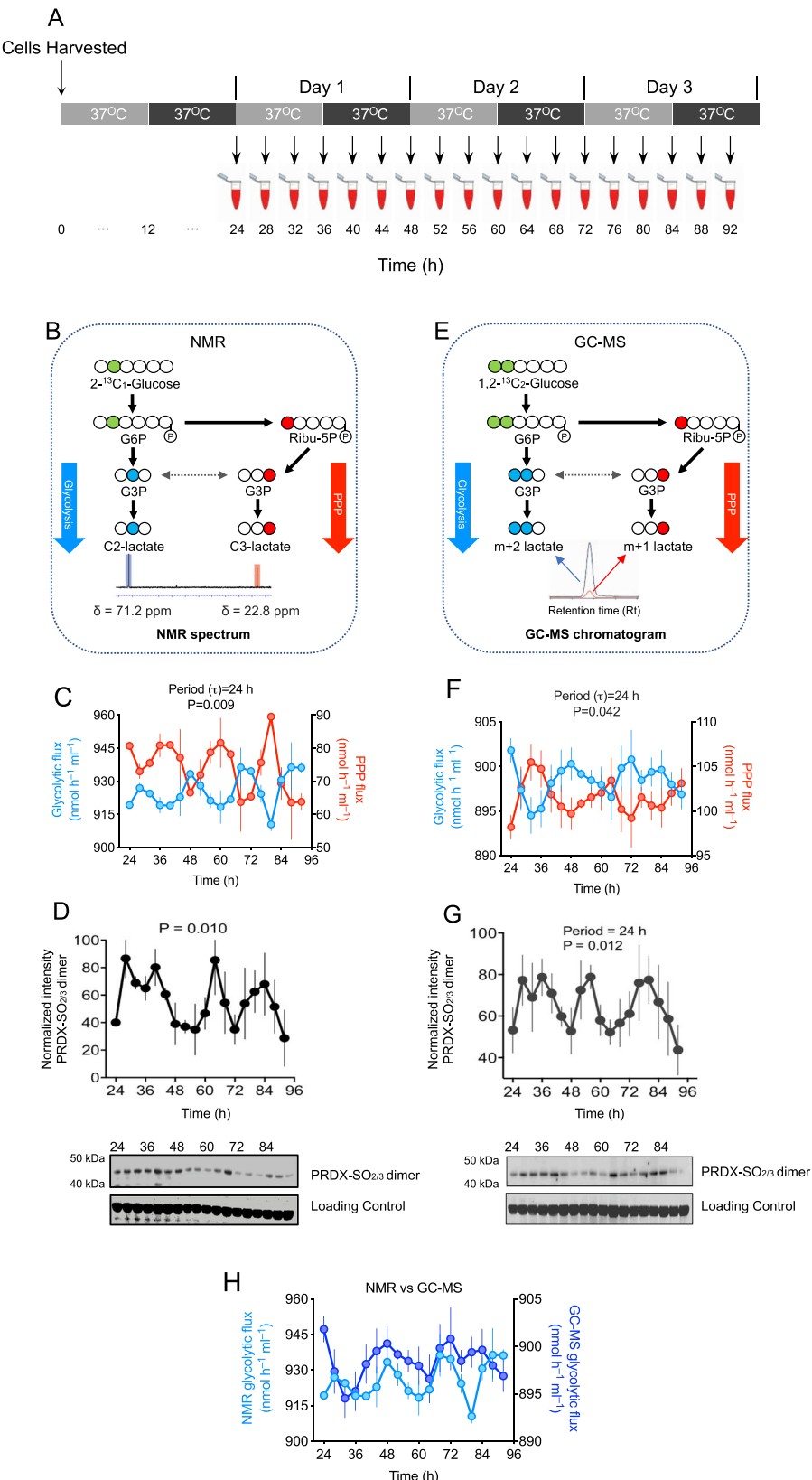

specificity for glycolysis and PPP. Furthermore, treatment with either pathway inhibitor abolished of time of day variation in metabolites (Fig. 5E). Similarly, temporal variation of ATP (which reflects glycolytic energy output) was abolished by HA, and analogously 6AN had an identical effect on oxidized glutathione (an output of the PPP) (Fig. 5F). Together, these results

demonstrate that there is tight coupling between core glucose metabolism and peroxiredoxin oxidation rhythms in RBCs, and that perturbing flux through glucose-metabolizing pathways has a potent effect on redox rhythms.

In addition to PRDX oxidation rhythms in RBCs, we also identified 24-h oscillations of the redox coenzymes NADH and

**Fig. 3 Circadian regulation of glycolysis and pentose phosphate pathways (PPP) fluxes in human red blood cells (RBCs). A** Schematic showing experimental protocol used to collect samples. RBCs from $n = 3$ human subjects were incubated with either 11 mM 2-$^{13}C_1$-glucose or 1,2-$^{13}C_2$-glucose and kept under constant conditions (37 °C in continuous darkness) and sampling was performed every 4 h over 3 days period. **B** Schematic showing fate of 2-$^{13}C_1$-Glucose metabolized through glycolysis and the pentose phosphate pathway (PPP). Metabolic fluxes for glycolysis and PPP were quantified by $^{13}C$-NMR (nuclear magnetic resonance) spectroscopy. When 2-$^{13}C_1$-glucose is used as a tracer, it is metabolized via glycolysis and produces singly labeled $^{13}C_1$-2-lactate with chemical shift, $\delta = 71.2$. The same 2-$^{13}C_1$-glucose metabolized through the PPP produces singly labeled $^{13}C_1$-3-lactate with chemical shift $\delta = 22.8$. **C** Rhythmic glycolytic and PPP fluxes ($P = 0.009$) in RBCs measured by NMR. P-values were obtained from rhythmicity analysis using RAIN algorithm. Graph bars present mean ± s.e.m ($n = 3$ biological replicates). **D** Rhythmic PRDX oxidation used as a control for circadian rhythmicity in RBCs for measuring metabolic fluxes by NMR experiment. P-values were obtained from rhythmicity analysis using RAIN algorithm. Data are presented mean ± s.e.m ($n = 3$ biological replicates). **E** Schematic showing fate of 1,2-$^{13}C_2$-Glucose metabolized by glycolysis and the pentose phosphate pathway (PPP). Metabolic fluxes were measured with GC-MS. When 1,2-$^{13}C_2$-glucose is used as a tracer, it is metabolized via glycolysis and produces doubly labeled 2,3-$^{13}C_2$-lactate. The same 1,2-$^{13}C_2$-glucose metabolized through the PPP produces singly labeled $^{13}C_1$-lactate. The glycolytic m + 2 isotopomer of $^{13}C_2$-lactate and the pentose phosphate pathway m + 1 isotopomer of $^{13}C_1$-lactate were thus used for flux measurements (see Methods). **f** Rhythmic regulation of glycolysis and PPP fluxes in RBCs measured by GC-MS. P-values were obtained from rhythmicity analysis using RAIN algorithm. Data are presented mean ± s.e.m ($n = 3$ biological replicates). **G** Rhythmic PRDX oxidation used as a control for circadian rhythmicity in RBCs for measuring metabolic fluxes by GC-MS experiment. P-values were obtained from rhythmicity analysis using RAIN algorithm. Data are presented mean ± s.e.m ($n = 3$ biological replicates). **H** Comparison of glycolytic flux measurements in RBCs measured by NMR and GC-MS methods. This correlation figure is obtained from Fig. 3C, F. Data are presented mean ± s.e.m ($n = 3$ biological replicates).

NADPH previously[11]. Since RBCs do not possess organelles, such as mitochondria, NADH and NADPH originate from glycolysis and PPP[29]. The redox coenzymes NADH and NADPH produced by glycolysis and PPP are likely links between glucose metabolism and redox cycling of peroxiredoxins. However, their respective role in the generation of circadian rhythmicity in RBCs remains undefined.

We first measured NADH and NADPH in RBCs every 4 h over a period of 24 h after treating cells with the glycolytic and PPP inhibitors, HA and 6AN, respectively (Fig. 6A). We did not find discernible 24-h rhythms in either NADH or NADPH when cells were treated with the inhibitors (Fig. 6B, C), although a residual non-circadian (12-h) NADH rhythm appeared to persist in cells treated with HA (Fig. 6B). This suggests coupling between cellular metabolism and redox factors NAHD and NADPH. Thus, rhythmic glucose flux is necessary for robust redox oscillations (PRDX-SO$_{2/3}$, NADH, and NADPH). To test whether the converse is true, that redox oscillations are needed for flux rhythms, we treated RBCs with a compound that inhibits PRDX oxidation (conoidin A)[30–33] and one that abolishes PRDX oxidation rhythms (MG-132)[17] in RBCs (Fig. 6D). Treatment with either inhibitor resulted in non-rhythmic flux (Fig. 6E). These data indicate that rhythmic glucose metabolism is a crucial determinant of circadian redox rhythms, and reciprocally, redox rhythms are required for flux rhythms in RBCs.

**Metabolic flux rhythms in nucleated mammalian cells**. We next tested whether metabolic flux oscillations also occur in nucleated cells. Human U2OS cells have been extensively used to investigate molecular clock mechanisms, including small molecule screening[28,34,35]. Unlike RBCs, U2OS cells exhibit autonomous circadian rhythms in gene transcription and translation. U2OS cells also exhibit circadian PRDX oxidation rhythms[28] and these redox rhythms are intrinsically coupled to clock components through reversible redox modifications[36]. As well as being essential redox factors in RBCs, NADH and NADPH influence the activity of transcription factors in nucleated cells by modulating the binding of circadian components such as CLOCK and BMAL1 to the genome[37,38].

We therefore determined glycolytic and PPP fluxes in U2OS cells. Cells were grown to confluence (i.e., non-dividing to exclude contributions from the cell cycle), cultured for three days in standard glucose (25 mM) medium, and then synchronized with a 15 min 100 nM dexamethasone pulse[28]. Then, we replaced medium with that containing 25 mM 1,2-$^{13}C_2$-glucose. Beginning

24 h after synchronization, we took samples for 48 h and measured metabolic flux (Fig. 7A). GC-MS analysis of cellular extracts showed that glycolytic and PPP flux exhibit circadian oscillations, as in RBCs ($P < 0.001$ by RAIN; Fig. 7B; only glycolytic flux shown for clarity).

**Glycolytic and PPP fluxes are independent of BMAL1 in nucleated cells**. Metabolic regulation is closely associated with the circadian clock machinery, and REV-ERB proteins play critical roles in feedback regulation of the core TTFL oscillator because they are direct targets of BMAL1 and CLOCK transcription factors[39]. Accordingly, recently developed REV-ERB agonists, such as SR9011, alter circadian behavior and metabolism in mice by affecting core clock genes[40]. We treated U2OS cells with 50 μM SR9011, which resulted in severe blunting of circadian oscillations of the reporter *Bmal1:luciferase* (*Bmal1:luc*). The effect was reversible on washing out the compound (Fig. 7C, Supplementary Fig. 9). Of note, although higher concentrations of SR9011 (e.g., 100 μM) completely abolished rhythms, recovery of rhythms was not apparent on wash out because of cell death at higher concentrations (Supplementary Fig. 9). Despite compromised transcriptional rhythms in SR9011-treated cells, we found persistent circadian rhythms of glycolytic and PPP fluxes (Fig. 7D; only glycolytic flux shown for clarity).

To validate these results in a different model, we performed time courses on skin fibroblasts from adult *Bmal1*$^{−/−}$ and *Bmal1*$^{+/+}$ mice. BMAL1 (MOP3) is considered to be an essential component of the circadian transcriptional clockwork, and is necessary for behavioral rhythms[41]. *Bmal1*$^{+/+}$ and *Bmal1*$^{−/−}$ fibroblasts were grown to confluence, cultured for three days in standard glucose (25 mM) medium, and then synchronized for 15 min with a 100 nM dexamethasone pulse[28,42]. We then replaced medium with that containing 25 mM 1,2-$^{13}C_2$-glucose (Fig. 7E). Twenty-four hours after synchronization, we took samples every three hours for 48 h and measured metabolic flux (Fig. 7E). We found robust rhythms in relative glucose flux in both *Bmal1*$^{+/+}$ and *Bmal1*$^{−/−}$ cells (Fig. 7F). These findings are consistent with data suggesting residual, but non-circadian, oscillations of glycolytic metabolites (biphosphoglycerate, ATP, ADP, and lactate) in U2OS cells when *Bmal1* was knocked down by RNA interference (RNAi)[35]. Moreover, non-canonical circadian mRNA, protein and post-translational rhythms (including PRDX-SO$_{2/3}$) were also reported in *Bmal1*$^{−/−}$ cells and tissue[42], showing that metabolic flux oscillations are an additional type of BMAL1-independent circadian rhythm.

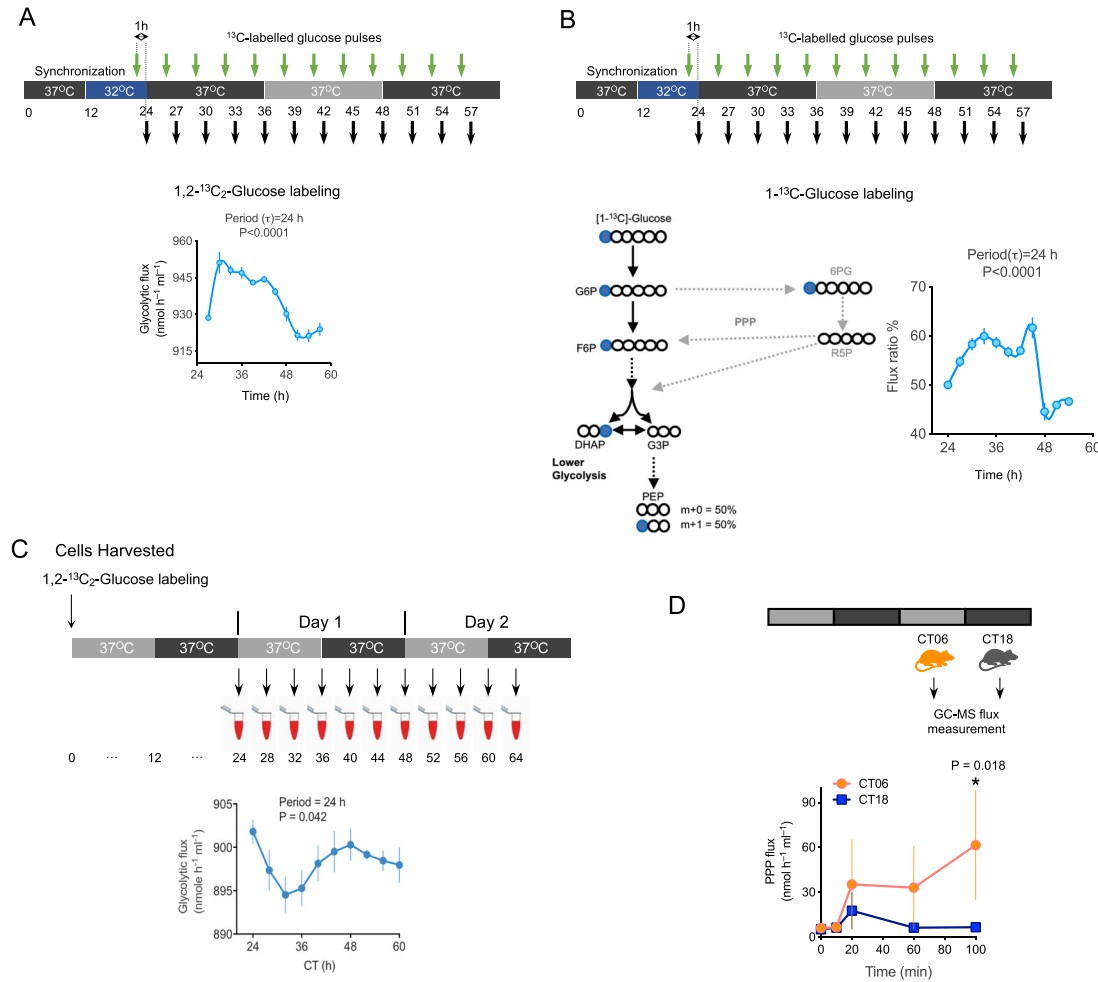

**Fig. 4 Rhythmic metabolic fluxes in human RBCs subjected to dynamic glucose labeling. A** Schematic showing the sampling protocol for dynamic glucose pulse labeling and below calculated fluxes from these labeling experiments. Oscillations of glycolytic fluxes using either 1,2-$^{13}C_2$-glucose labeling are shown below the schematic. Data for 1,2-$^{13}C_2$-glucose flux are mean ± s.e.m ($n = 3$ biological replicates). Samples were entrained with 12-h: 12-h 37 °C: 32 °C and maintained at constant conditions. **B** Independent validation of metabolic flux with isotope 1-$^{13}C1$-Glucose. Schematic of 1-$^{13}C1$-Glucose labeling and calculation of PEP isotopomer ratio to calculate glycolytic flux ratio. G6P = Glucose-6-phosphate, G3P = Glyceraldehyde 3-phosphate, F6P = Fructose-6-phosphate, DHAP = Dihydroxyacetone phosphate, PEP = Phosphoenolpyruvate, R5P = Ribose-5-phosphate, 6PG = 6-Phosphogluconate. Glycolytic (fgly) flux ratio for 1-$^{13}C_1$-glucose experiments was determined using the mass isotopomer distribution of Phosphoenolpyruvate (PEP). Data for 1-$^{13}C_1$-glucose flux ratio are mean ± s.e.m ($n = 4$ biological replicates). *P*-values report significance by ANOVA (effect of time). **C** Schematic showing the sampling protocol for calculated fluxes from 1,2-$^{13}C_2$-glucose labeling experiments. Oscillations of glycolytic fluxes are shown below the schematic. Data for 1,2-$^{13}C_2$-glucose flux are mean ± s.e.m ($n = 3$ biological replicates). *P*-values were obtained from rhythmicity analysis using RAIN algorithm Note these data are replotted from Fig. 3F to compare phase of glycolytic flux to Fig. 4A. **D** Measurement of RBC flux in circulating mouse RBCs at opposite circadian phases. Mouse RBCs ($n = 3$ biological replicates) were harvested at opposite phases of the circadian cycle (see schematic): CT06 and CT18 (CT, circadian time: subjective day CT00-CT12; subjective night CT12-24). *P*-values report significance by one way ANOVA.

## Discussion

Although gene expression cycles are essential for the temporal coordination of physiology, we and others have previously shown that rhythms in redox balance is a cell-intrinsic phenomenon that persists without any gene expression cycles in non-nucleated mammalian red blood cells[11,17]. This circadian rhythm in redox balance is seen across the domains of life, and cells have evolved a number of defense mechanisms to counteract the deleterious consequences of oxidative stress, including the use of peroxiredoxin proteins[10,43,44]. Among these processes, metabolism is a key player, and is also evolutionarily conserved to maintain cellular energy and survival[45]. However, how cytosolic rhythmic redox balance interacts with metabolism has not been elucidated. At different levels of biological organization, from the whole body to single cells, a significant portion of metabolism is under circadian control, leading to the prevailing view that biological

cycles drive metabolic rhythms[46,47]. Indeed, recent reports suggest rhythmic metabolites from lipid metabolism, and NAD+ biosynthesis from mitochondrial metabolism, are controlled by the clock[48].

However, such circadian control of metabolism cannot happen in human red blood cells, where there are no gene expression cycles and/or organelles such as mitochondria. We uncovered a circadian metabolome in human RBCs and also showed circadian rhythms of glycolysis and PPP metabolites, indicating that cellular metabolism may be an integral part of non-transcriptional circadian clocks. RBCs depend on glycolysis for ATP, and on the PPP to maintain the redox balance through NADPH[29]. Because metabolites are functionally dependent on input of metabolic fluxes through a number of enzyme pathways, the contribution of fluxes in generating cycling metabolites is important. Our labeling experiments with human RBCs revealed rhythmic regulation of

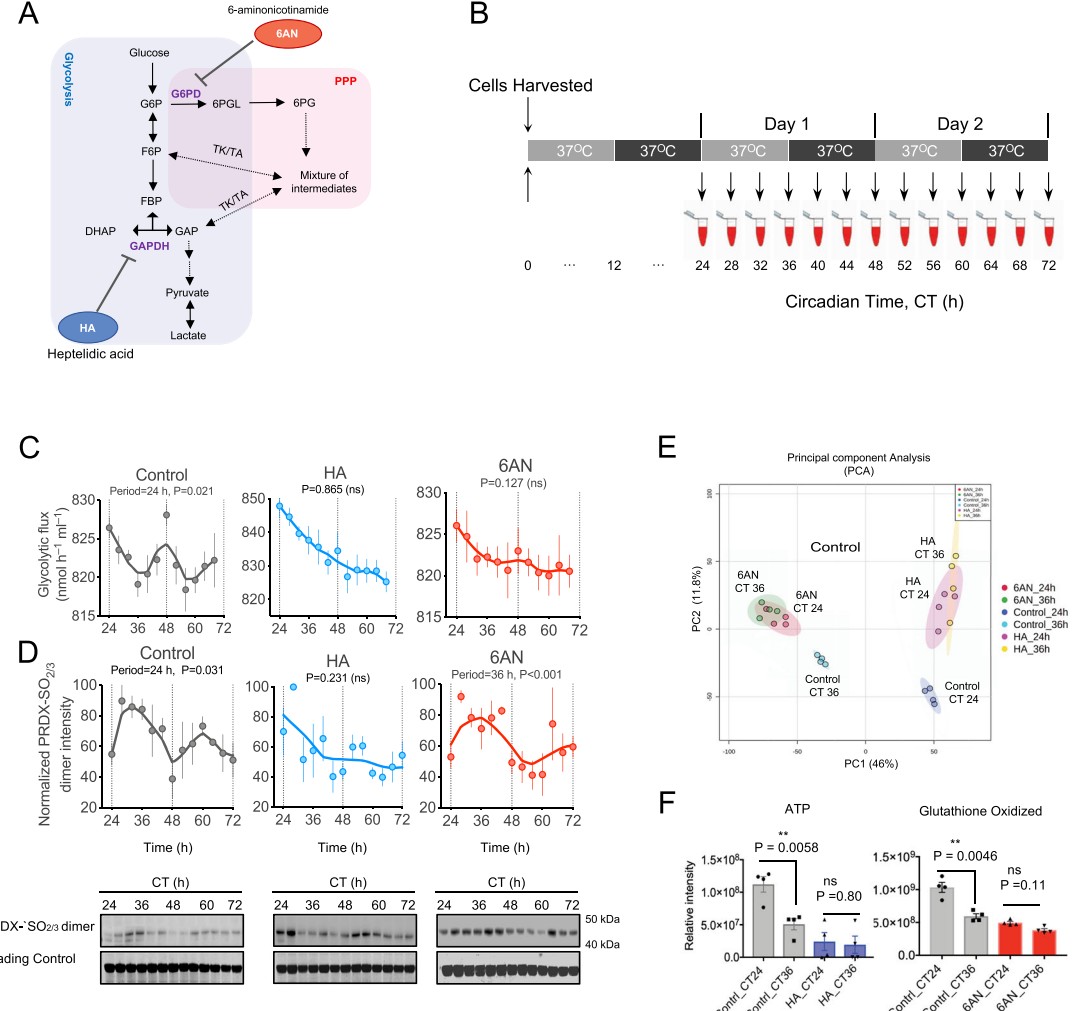

**Fig. 5 Metabolic inhibition abolishes circadian metabolic flux and PRDX oxidation rhythms in human red blood cells (RBCs). A** Schematic showing key steps and metabolites in glycolysis and the pentose phosphate pathway (PPP) and the points at which the glycolytic inhibitor, heptelidic acid (HA) and the PPP inhibitor, 6-aminonicotinamide (6AN), act in the pathways. G6P = Glucose-6-phosphate, F6P = Fructose-6-phosphate, FBP = Fructose-1,6-biphosphate, DHAP = Dihydroxyacetone phosphate, 6PGL = 6-Phosphogluconolactone, 6PG = 6-Phosphogluconate, GAP = Glyceraldehyde-3-phosphate, GAPDH = Glyceraldehyde 3-phosphate dehydrogenase, G6PD = Glucose-6-phosphate dehydrogenase, TK = Transketolase, TA = Transaldolase. **B** Schematic for sample collection. Freshly prepared human RBCs treated with metabolic inhibitors 1.9 μM HA, 10 mM 6AN, or vehicle control (DMSO) and then kept under constant conditions (37 °C, continuous darkness) and sampling was performed every 4 h. **C** Effect of inhibitors on metabolic fluxes through glycolysis measured by GC-MS. $P$-values were obtained from rhythmicity analysis using RAIN algorithm. Graph bars present mean ± s.e.m ($n = 4$ biological replicates). **D** The effect of metabolic inhibitors on peroxiredoxin (PRDX) oxidation in RBCs was measured by immunoblotting for PRDX-SO$_{2/3}$ dimer. Blots are shown with their respective loading controls (Coomassie blue gel images). Graph bars present mean ± s.e.m ($n = 3$ biological replicates). Full blots from all subjects are shown in Supplementary Fig. 7. $P$-values were obtained from rhythmicity analysis using RAIN algorithm. **E** Effect of HA and 6AN on the RBC metabolome. Principal Components Analysis (PCA) plot showing components 1 and 2 for all metabolomic samples showing good separation between control, HA and 6AN treated samples. Circadian time dependent changes were observed in control along PC1 (CT 24 vs. CT 36), while circadian time dependent changes are abolished in samples treated with metabolic inhibitors. Ellipses indicate the 95% confidence intervals of each grouping of samples on the plot. **F** Time-dependent changes in ATP and oxidized glutathione at CT24 and CT36. Controls show significant ($P < 0.01$) temporal variation of abundance of ATP and oxidized glutathione, while samples treated with metabolic inhibitors had no time-dependence (not significant, n.s). $P$-values report significance by one way ANOVA. Graph bars present mean ± s.e.m ($n = 4$ biological replicates).

glycolytic and PPP flux. These fluxes have opposite phases, with the PPP reaching its peak during the day, aligned with peroxiredoxin peak oxidation, while glycolysis is active during the night (Supplementary Fig. 10).

It is likely that RBC redox rhythms are synchronized to human physiology through blood oxygen levels. Indeed, the latter exhibit a 24-h pattern, and oxygen levels reach their maximum in the day (during biophysical peak activity), and decrease to their lowest levels during at night[49]. This leads to greater oxygenation of hemoglobin in RBCs, and the generated ROS are likely to entrain

cellular redox state and oxidation of peroxiredoxin to synchronize the RBC 24-h oscillations in oxidative stress to body physiology[11,17]. To counteract cellular oxidative damage, rhythmic NADPH production by the PPP may thus be required to prevent cellular damage caused by the daily auto-oxidation of hemoglobin[50–52].

Our results show that during the circadian night, when oxygen is lowest, glycolysis reaches its peak. Active dynamic re-routing of carbohydrate flux is key to counteracting oxidative stress, and our results indicate that switching metabolic flux through PPP and

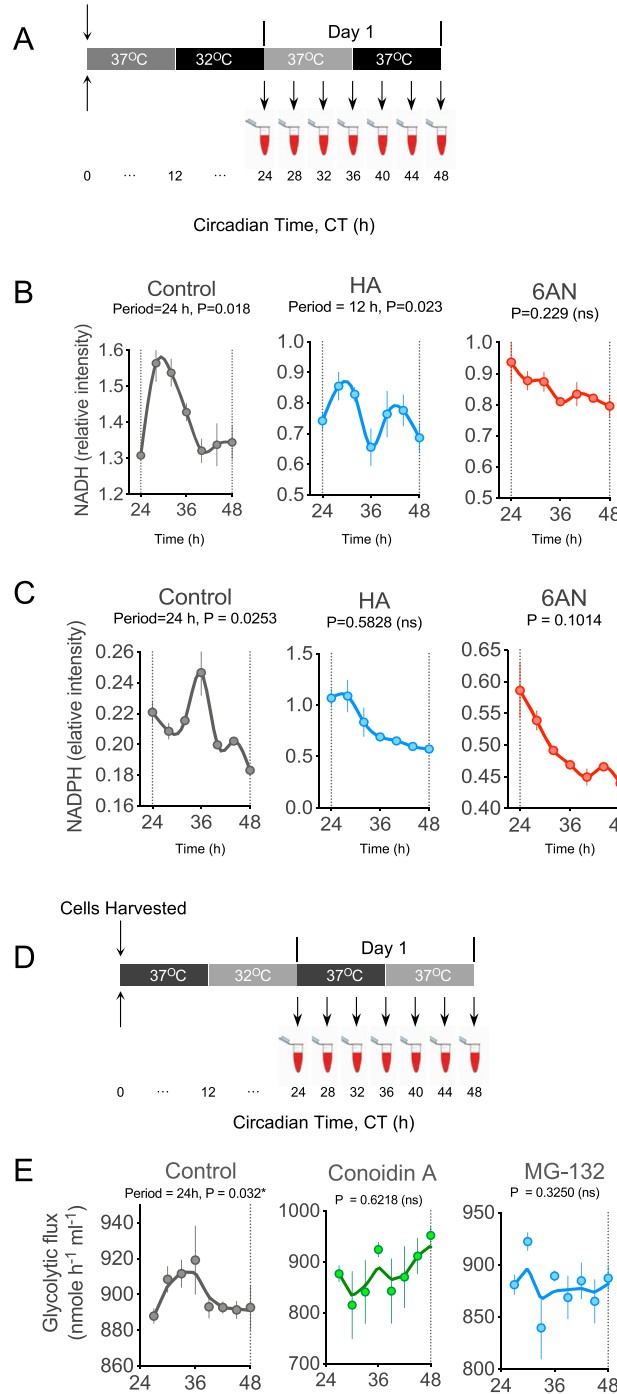

**Fig. 6 Metabolic inhibition abolishes rhythmic redox cofactor accumulation in human red blood cells (RBCs). A** Schematic showing experimental protocol used to collect samples. RBCs from $n = 3$–$4$ human subjects were incubated with 11 mM 1, 2-13C2-glucose and kept under constant conditions (37 °C in continuous darkness) and sampling was performed every 4 h. Cells were treated with metabolic inhibitors, HA and 6AN at the starting of the experiment. **B** Effect of metabolic inhibitors HA and 6AN on the rhythmicity of the redox coenzyme, NADH. Graph bars present mean ± s.e.m ($n = 6$ biological replicates) **C** Effect of metabolic inhibitors HA and 6AN on the rhythmicity of the redox coenzyme, NADPH. Graph bars present mean ± s.e.m ($n = 6$ biological replicates). **D** Schematic showing experimental protocol used to collect samples after treating with inhibitors Conoidin A and MG-132. **E** RBCs were treated with inhibitors of PRDX oxidation (Conoidin A) or PRDX oxidation rhythms (MG-132) and their effect on glucose flux was assessed. All data are mean ± s.e.m. ($n = 4$ biological replicates). The 24-h rhythmicity $P$-value (determined by RAIN) for each profile is shown with each plot, in addition to the best fit period of the rhythm if this was not 24 h. ns, not significant ($P > 0.05$).

Our experiments with U2OS cells reveal rhythmic regulation of glycolysis and PPP fluxes in nucleated human cells. Furthermore, experiments using *Bmal1* knockout mouse cells show that circadian flux rhythms persist in the absence of a functional TTFL network. Indeed, glycolysis and PPP fluxes have higher amplitudes in comparison to control (*Bmal1*[+/+]) cells. We recently reported circadian oscillations of the transcriptome, proteome and phosphoproteome of *Bmal1* knockout mouse cells and tissue[42]. The flux data in *Bmal1* knockout cells presented in this study thus strongly support the notion of TTFL-independent circadian timekeeping mechanisms. Given that metabolic oscillations are present in both nucleated cells genetically lacking a functional TTFL[53], and in the complete absence of transcription or translation (as shown here), the most parsimonious explanation is that rhythmic metabolism is at the core of circadian timekeeping. Thus, our work paves the way to explore the role of metabolic components in regulating circadian rhythms in diverse model systems.

## Methods

### Resource availability
*Contact and materials availability*. Further information and requests for resources and reagents should be directed and will be fulfilled by the senior author, Akhilesh B Reddy (areddy@cantab.net). This study did not generate new unique reagents.

### Experimental model and subject details
*Human participants and ethics statement*. Studies were conducted in accordance with the principles of the Declaration of Helsinki, with approval from the Health Research Authority's (UK) Research Ethics Committee (Reference number 12/EE/0370) and local ethical approval by The Francis Crick Institute's Ethics Review Board. The Francis Crick Institute is licensed by the Human Tissue Authority (HTA) to store human samples for the purposes of research (License number 12650). The research complies with all requirements of the relevant HTA Code of Practice. All volunteers provided written informed consent after receiving a participation information sheet containing detailed information of the study procedures. Participants were screened for self-reported health issues including sleep disorders, night shift work, and regular high alcohol consumption, and excluded if they met any of these criteria. Sample size for all experiments mentioned at corresponding experimental figure legends.

*Human red blood cells (RBC) culture*. Fresh blood samples (9–10 ml) were collected from each healthy volunteer in 10 ml tubes containing trisodium citrate (Sarstedt, S-Monovette 02.1067.001). RBCs from each donor were separately isolated using 5 ml of Histopaque (Histopaque-1007, density 1.077 g/ml, sigma) by density gradient centrifugation for 15 min at $700 \times g$. The obtained RBC pellet was washed three times using 10 ml sterile PBS (Sigma-Aldrich). The washed pellet was diluted to 9 ml with Krebs–Henseleit Buffer (290 mOsm, pH 7.40) supplemented with 100 U/ml penicillin and 100 µg/ml streptomycin (Sigma-Aldrich) and 0.1% BSA (Sigma-Aldrich). 500 µl of anti-CD15 Dynabeads (Life Technologies 11137D) were added to each tube to remove any remaining nucleated cells, which were extracted

glycolysis over the circadian cycle could fulfill this goal (Supplementary Fig. 10). GAPDH functions as a metabolic switch for re-routing carbohydrate[52]. We have shown that inhibiting GAPDH results in arrhythmic metabolic fluxes, which leads to arrhythmic redox balance as shown by flat NADPH, NADH, and peroxiredoxin oxidation profiles. Thus, our data suggest daily cross-talk of glucose metabolism and redox factors is required to maintain circadian oscillations in human RBCs.

In eukaryotes, prevailing clock models revolve around transcription-translation feedback loops (TTFLs)[6]. How cell metabolism couples to rhythmic peroxiredoxin state and whether these metabolic rhythms are independent of TTFL was unclear.

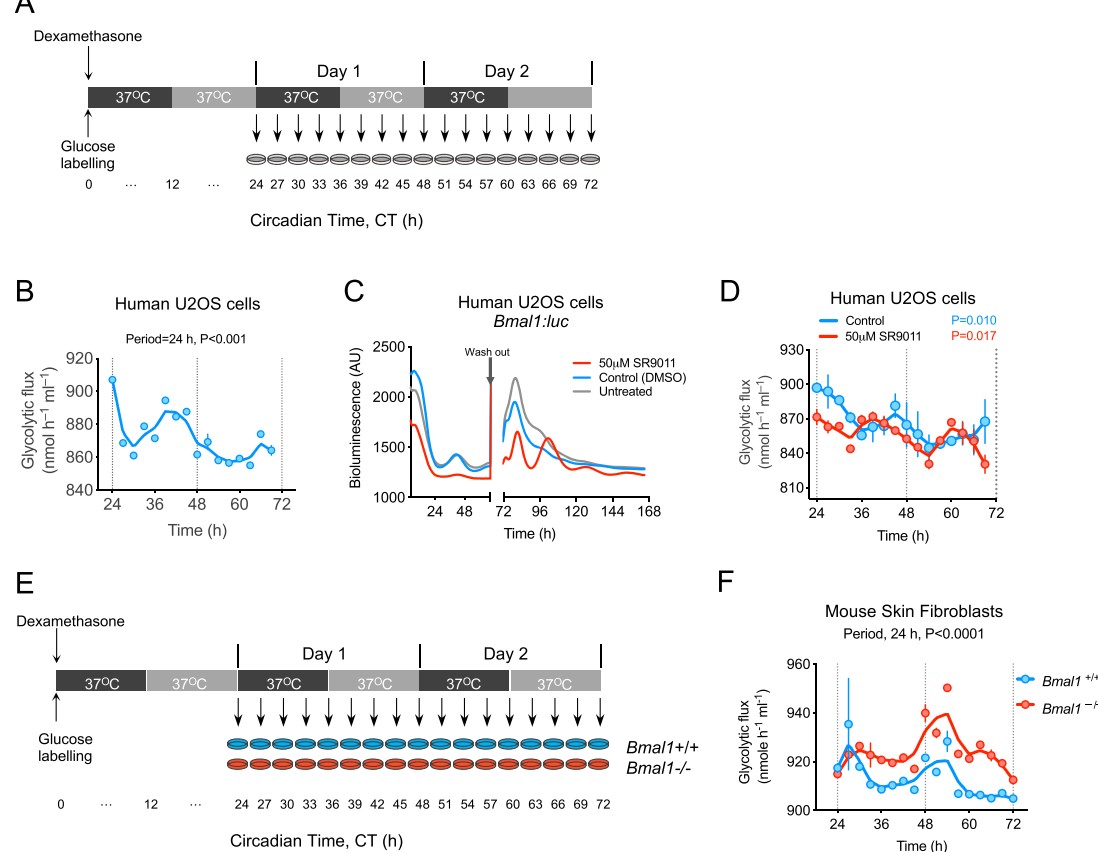

**Fig. 7 Circadian regulation of metabolic flux in nucleated cells. A** Human U2OS cells were synchronized with a 15 min dexamethasone pulse treatment and their medium replaced with DMEM medium containing 25 mM 1,2-$^{13}$C$_2$-glucose. They were then kept under constant conditions (37 °C, continuous darkness) and sampled every 3 h, starting 24 h after the dexamethasone pulse (and glucose labeling). **B** Glycolytic and pentose phosphate pathway (PPP) flux in U2OS cells was measured by GC-MS using the glycolytic m + 2 isotopomer of $^{13}$C$_2$-lactate and the pentose phosphate pathway m + 1 isotopomer of $^{13}$C$_1$-lactate (see "Methods" section). Data are mean ± s.e.m. (n = 3 biological replicates). The 24-h rhythmicity P-value (determined by RAIN) is shown with the plot. **C** Treatment of U2OS cells with 50 μM of SR9011 resulted in severe damping of bioluminescence rhythms that was reversible on drug wash out (indicated by arrow). Traces for cells treated with vehicle control (DMSO), and untreated cells, are also shown. Data are mean of n = 4 biological replicates (see Supplementary Fig. 9 for traces with error boundaries). **D** Metabolic flux in cells treated with SR9011 (or vehicle control). P-values were obtained from rhythmicity analysis using RAIN algorithm. **E** Adult skin fibroblasts from Bmal1$^{+/+}$ and Bmal1$^{-/-}$ mice were synchronized for a 15 min with dexamethasone and their medium replaced with DMEM medium containing 25 mM 1,2-$^{13}$C$_2$-glucose. Cells were then kept under constant conditions (37 °C, continuous darkness) and sampled every 3 h, starting 24 h after the dexamethasone pulse (and glucose labeling. **F** Glycolytic flux in adult skin fibroblasts from Bmal1$^{+/+}$ and Bmal1$^{-/-}$ mice. Metabolic flux measured by GC-MS. Flux data are mean ± s.e.m. (n = 3 biological replicates).

after 15 min incubation with the aid of a magnet (Life Technologies). Krebs-Henseleit Buffer containing 11 mM D-Glucose (Sigma-Aldrich K3753) was used for culturing RBC and un-targeted metabolite profiling experiments. For metabolic labeling experiments, unlabeled D-glucose was replaced with either 11 mM U-$^{13}$C$_6$-D-glucose (99%, CLM-1396, CK Isotopes Ltd) or 2-$^{13}$C$_1$-D-glucose (99%, CLM-504, CK Isotopes Ltd). Different volumes of purified RBCs were used for untargeted metabolite analysis (1 ml of RBC per sample), metabolic flux experiments with NMR (1 ml of RBC per sample) and GC-MS (200 µl of RBC per sample). RBC samples were maintained at 37 °C in a microprocessor-controlled incubator (Eppendorf Galaxy 170 R) in complete darkness, in sealed tubes, for sampling. For metabolic inhibition experiments, heptelidic acid was dissolved in DMSO to make a stock solution. Compounds were added in Krebs-Henseleit Buffer to their final concentration. Drugs were added to RBCs to reach a final concentration of 10 mM 6-Aminonicotinamide (CAS 329-89-5, Sigma A68203), or 1.9 µM heptelidic acid (CAS 74310-84-2, Biovision 2215–1000), or 3 µM MG-132, or 5 µM conoidin A. Control samples were treated with 0.5% DMSO.

*Experiments with U2OS cells.* Stably-transfected *Bmal1:luc* U2OS cells were a kind gift of Dr Andrew Liu (University of Memphis). Cells were cultured in a humidified 5% CO$_2$ incubator in Dulbecco's Modified Eagle's medium (DMEM) (Sigma D6546) containing 4.5 g/l glucose, 1× Glutamax (Life Technologies 35050-038, 10% Newborn Calf Serum (Sigma 12023 C), 100 U/ml penicillin and 100 µg/ml strep-tomycin (Sigma P0781), 1× MycoZap Plus-PR (Lonza) and blasticidin 2 µg/ml. For metabolic flux experiments, unlabeled glucose was replaced with 25 mM 1,2-$^{13}$C$_2$-D-glucose (99%, CLM-504, CK Isotopes Ltd). For bioluminescence recordings,

U2OS cells were grown to confluence in 96-well plates in the above medium and synchronized by changing the medium to Air Medium[28,54]: DMEM (Sigma D5030) supplemented with 4.5 g/L glucose (Sigma G8644), 1× Glutamax (Life Technologies 35050-038), 10% Newborn Calf Serum (Sigma 12023 C), 100 U penicillin/ml and 100 µg/ml streptomycin, 0.5× B-27® Supplement (Life Technologies 17504-044), 20 mM HEPES (Sigma H0887), 1 mM Luciferin (Biosynth L8220), 1× Non-Essential Amino acids (Sigma M7145), 0.035% NaHCO$_3$ (Sigma S8761), 1× MycoZap Plus-PR (Lonza) and blasticidin 2 µg/ml. For experiments using the REV-ERB agonist SR9011 (Caymen Chemical 11930; CAS 1379686-29-9), the compound was solubilized in DMSO and diluted in Air Medium to a final concentration of 50 or 100 µM. Control cells were treated with a matched concentration of DMSO. Bioluminescence assays were conducted in custom-made light-tight Alligator bioluminescence recording systems (Carin Research Ltd, Faversham, UK) composed of a CCD camera (Andor iKon-M 934) placed on the top of Galaxy 170 R incubator (Eppendorf). All bioluminescence experiments were performed at 37 °C in darkness. Plate luminescence images were captured every 30 min over seven days.

*Experiments with mice.* Animal studies were carried out in concordance with an approved protocol from Institutional Animal Care and Use Committee (IACUC) at Perelman School of Medicine at the University of Pennsylvania, or under license by the United Kingdom Home Office under the Animals (Scientific Procedures) Act 1986, with Local Ethical Review by the Francis Crick Institute Animal Welfare & Ethical Review Body Standing Committee (AWERB). Male C57BL/6J mice, 8–10 weeks old, were purchased from Charles River and allowed to acclimatize and

entrain in a 12-h light-dark cycle (LD). Light intensity during light and dark periods was 200 and <3 lux, respectively. Humidity and temperature ($21 \pm 1\,°C$) were kept within standard ranges. After 3 weeks, mice were transferred to constant darkness (DD; dim red light, <3 lux) and on the second day of DD blood was collected by cardiac puncture under terminal anesthesia (sodium pentobarbital, 170 mg/kg, intraperitoneal) at CT24 and CT36.

*Experiments with* Bmal1$^{-/-}$ *mouse skin fibroblasts.* Bmal1$^{+/+}$ and Bmal1$^{-/-}$ adult mouse skin fibroblasts (MSF) cells were grown in Dulbecco's Modified Eagle Medium (DMEM) containing 10% (v/v) HyClone III Serum (Analab; Cat # SH30109.03), 1/100 Glutamax-I (Invitrogen; Cat # 35050-038), 1/100 Penicillin-Streptomycin (SIGMA; Cat # P4333) and 1/500 MycoZap™ (Lonza; Cat # VZA-2022) in multiple six-well plates until fully confluent ($n = 3$, per time-point, per genotype). Confluent MSF cells were treated with 100 nM (final concentration) of dexamethasone (DEX) for 15 min to synchronize the cells. MSF cells were then washed three times with PBS ($37\,°C$) and were incubated in HEPES-buffered Medium; 1× DMEM powder (SIGMA; Cat # D5030), 5 mg/ml 1, 2-13C2 (99%) D-glucose (Cambridge Isotope Laboratories, Cat # CLM-504-1), 0.35 mg/mL sodium bicarbonate, 0.01 M HEPES, 5% (v/v) HyClone III Serum, 1/100 Glutamax-I, 1× B-27 supplement (LifeTech Cat # 17504-044), 1× Non-Essential Amino acids, and 1/500 MycoZap™, pH 7.4 (adjusted with HCl) and osmolality 350 mOsm (adjusted with NaCl) at $37\,°C$ under DD cycle. Twenty-four hours after the DEX treatment, MSF cells were harvested at every three-hour interval for two days for subsequent metabolic flux analysis.

**Metabolite extraction for untargeted metabolite profiling.** At each time point, 1.5 ml Eppendorf tubes containing 1 ml of purified RBCs in Krebs-Henseleit Buffer were collected from the incubator and samples were immediately centrifuged for 2 min at $375 \times g$ at $4\,°C$. The supernatant was removed and the RBC pellet washed twice with ice-cold PBS. RBC metabolite extractions were performed using a reported protocol[55] with slight modifications. Briefly, metabolites were extracted from RBCs by adding 450 µl of methanol. Vortex mixed for 10 s to lyse the cells in methanol. Immediately, 200 µl of chloroform and 200 µl of water was added to the pellet at $4\,°C$, followed by sonication for 10 min and then vortexing for 20 min. Samples were then centrifuged at $18,400 \times g$ for 20 min at $4\,°C$. The upper aqueous layer was collected and lower chloroform layer (containing non-polar metabolites) was discarded. Two more extractions were performed on the same RBC pellet with 50% methanol:water with vortexing for 20 min, and centrifugation at $18,400 \times g$ for 20 min at $4\,°C$. 13C$_5$-15N-Valine was used as an internal standard during all extractions. The three extracts were combined and dried with a vacuum concentrator (Concentrator plus, Eppendorf). The dried extracts were subsequently used for GC-MS and LC-MS analysis for metabolite profiling.

**UPLC-MS based data acquisition for untargeted metabolite profiling.** Dried samples were reconstituted in 500 µl methanol:water (1:1, v/v). LC-MS analysis was conducted using a Dionex UltiMate Liquid Chromatography (LC) system coupled to a Q-Exactive Orbitrap mass spectrometer (both Thermo Scientific), adapted from a reported method[56]. LC separation was performed using hydrophilic interaction chromatography (HILIC) on a ZIC-pHILIC column (150 mm×4.6 mm, 5 µm particle size; Merck Sequant) with a gradient solvent A (20 mM ammonium carbonate in water; Optima HPLC grade, Sigma Aldrich) and solvent B (acetonitrile; Optima HPLC grade, Sigma Aldrich). A 15 min elution gradient of 80 to 20% Solvent B was used, followed by a 5 min wash of 5% Solvent B and 5 min re-equilibration. Other LC parameters were: flow rate 300 µl/min; column temperature $25\,°C$; injection volume 10 µl; auto sampler temperature $4\,°C$. MS was performed with positive/negative polarity switching using a HESI II probe. MS parameters were: spray voltage 3.5 kV and 3.2 kV for positive and negative modes, respectively; probe temperature $320\,°C$; sheath and auxiliary gases were 30 and 5 arbitrary units, respectively; full scan range: 70 to 1050 $m/z$ with settings of AGC target and resolution as Balanced and High ($3 \times 10^6$ and 70,000) respectively. Data was recorded using Xcalibur 3.0.63 software (Thermo Scientific). To enhance calibration stability, lock-mass correction was also applied to each analytical run using ubiquitous low-mass contaminants. Parallel reaction monitoring (PRM) parameters: resolution 17,500; collision energies were set individually in high-energy collisional dissociation (HCD) mode.

**GC-MS based data acquisition for untargeted metabolite profiling.** Dried sample extracts were used for GC-MS analysis. Untargeted metabolite profiling was performed by GC-MS using an Agilent 7890A-5975C GC-MSD after derivatization of metabolites with methoxyamine hydrochloride (20 mg/ml in pyridine, both Sigma) and N,O-bis-(trimethylsilyl)trifluoroacetamide (containing 1% trimethyl-chlorosilane)[57]. GC separation was achieved using an Agilent DB-5 MS column (30 m × 0.25 mm × 0.5 µm). The GC oven temperature program was: 70 °C, 2 min hold; ramp 12.5 °C/min to 295 °C, 0 min hold; ramp 25 °C/min to 320 °C, 3 min hold. Other GC parameters were: injection volume 1 µl; inlet temperature 270 °C; Helium was used as a carrier gas at a flow rate of 0.9 ml/min; transfer line temperature 280 °C. Electron impact ionization was used for mass spectrometry detection with scan range $m/z$ 50–565.

**Metabolic flux measurements with NMR.** For RBC metabolic flux experiments, cultures were incubated with Krebs-Henseleit Buffer containing 11 mM 2-13C$_1$-D-glucose and at each time point a 1.5 ml Eppendorf tube containing 1 ml RBCs was collected from the incubator, and immediately centrifuged for 2 min at $4\,°C$ at $375 \times g$. The supernatant was collected and the pellet washed with ice-cold PBS two times and centrifuged again for 2 min at $4\,°C$ and at $375 \times g$. The RBC pellet and was then immediately flash frozen in liquid nitrogen. Frozen samples stored at $-80\,°C$ till the analysis. Frozen samples were thawed at $4\,°C$, and then placed in a boiling water bath for 9 min to lyse cells and halt enzymatic activity[23,58]. Boiled lysates were sonicated for 10 min at $4\,°C$ on ice, followed by vortex mixed for 10 min. Samples were then centrifuged for 20 min at $18,400 \times g$ at $4\,°C$. Nine-hundred microliter of each supernatant was dried in a vacuum concentrator (Concentrator plus, Eppendorf). Dried sample extracts were suspended in 750 µl D$_2$O (Sigma 151882) containing 0.05% of 3-(trimethlysilyl)-1-propanesulfonicacid-d$_6$ sodium salt (Sigma 613150). 13C-NMR spectroscopy was performed on a Bruker AM-500 MHz spectrometer (Chemistry Department, University of Cambridge) or a Bruker Avance III 600 MHz spectrometer equipped with a 5 mm TCI Cryoprobe (MRC Biomedical NMR Centre, Francis Crick Institute). Spectra were recorded using a 30° 13C excitation pulse, 1 s acquisition time, and 3 s relaxation delay. The spectra were 1H-broadband de-coupled using WALTZ16, which was also employed during the relaxation delay to exploit the {1H}13C heteronuclear Overhauser enhancement. Labeled C2-lactate, C3-lactate and alpha-anomers, beta-anomers of glucose were identified and peak integrals were evaluated using the Bruker NMR software package TopSpin 3.5.

**Metabolic flux measurements with GC-MS.** For RBC metabolic flux experiments, cultures were incubated with Krebs–Henseleit Buffer containing 11 mM 1,2-13C$_2$-D-glucose and at each time point a 1.5 ml Eppendorf tube containing 200 µl RBCs was collected from the incubator, and immediately centrifuged for 2 min at $4\,°C$ at $375 \times g$. The supernatant was collected and the pellet washed with ice-cold PBS two times and centrifuged again for 2 min at $4\,°C$ and at $375 \times g$. The RBC pellet and was then immediately flash frozen in liquid nitrogen and metabolites extracted with 80% methanol:water followed by vortexing for 10 min, and then centrifugation at $18,400 \times g$ for 15 min at $4\,°C$. The aqueous layer was collected and dried using a vacuum concentrator (Concentrator plus, Eppendorf).

For U2OS cell experiments, cultures were incubated with medium containing 25 mM 1,2-13C$_2$-D-glucose and at each time point, culture supernatant was removed and cells washed with ice-cold PBS three times. Immediately, 1 ml of ice-cold 80% methanol:water (pre-cooled to $-80\,°C$) was added and cells collected by scraping into 1.5 ml Eppendorf tubes. Cells were flash frozen in liquid nitrogen and stored at $-80\,°C$.

For lactate isotopomer analysis, samples were derivatized as above and analyzed on an Agilent GC-MSD. Parameters were as for untargeted GC-MS analyses above, but with the following GC temperature gradient: 120 °C, 2 min hold; ramp 8 °C/min to 180 °C; ramp 20 °C/min to 290 °C, 3 min hold. Selective ion monitoring (SIM) for lactate isotopomers m + 0 (unlabeled 12C-lactate), m + 1 (13C$_1$-lactate), m + 2 (13C$_2$-lactate), m + 3 (13C$_3$-lactate).

Extracellular glucose from the media obtained from incubating RBCs with U-13C-Glucose over a period of 72 h were analyzed after drying of 1 µl of medium. Derivatization and GC-MS conditions were the same as mentioned above.

**Pulse labeling experiments.** Pulsed isotopic labeling was performed by feeding RBCs with either 1,2-13C$_2$-glucose or 1-13C-glucose. To minimize potential clock resetting and perturbation of cells due to replacement of normal glucose medium with labeled medium, we used conditioned medium, which was obtained by incubating RBCs for 1 h in labeled medium and storing immediately at $4\,°C$. Metabolic measurements for 1,2-13C2-glucose experiments were performed as follows. Briefly, 800 µl of the upper fraction was dried in a speed-vacuum for 4.5 h. The resulting dried sample was reconstituted in 200 µl of acetonitrile:water, vortexed for 20 s, and centrifuged at 13,300 rpm for 15 min. The undiluted sample was used for glucose measurements while a 30-fold diluted sample was utilized for lactate measurements. Samples were measured in analytical duplicates with each sample set run in a randomized manner with pooled quality control samples measured at the start of the run, after every 10th sample, and at the end of the run. For both glucose and lactate measurements, 2 µl of each sample was analyzed in a manner similar to the methods previously described[59,60], modified for isotopomer analysis. Targeted multiple reaction monitoring (MRM) methods were utilized to detect lactate and glucose isotopologues in each respective run. Data were processed and integrated using Waters TargetLynx software (version 4.1) with natural abundance correction and further processing of ion counts performed in R.

Metabolic measurements for 1-13C-glucose experiments was measured with Agilent LC-QTOF 6545. Metabolites were confirmed by running standards in parallel. Isotopic labeling was measured using Agilent MassHunter Profinder v. B.08.00 (Batch Isotopologue extraction) and the personal compound database and library (PCDL).

**Flux measurements in** Bmal1$^{-/-}$ **mouse skin fibroblasts.** MSF cells collected at each time point were washed with ice-cold PBS three times and then homogenized in 1 ml 80% methanol (precooled at $-80\,°C$). Samples were then flash frozen in

liquid nitrogen and were stored at −80 °C until extraction. In the extraction process, mild sonication was applied to the samples for 10 min (30 s on, 30 s off; medium power) using a Bioruptor Standard (Diagenode) instrument. Then the samples were vortexed for 20 mins at 4 °C and the lysates were centrifuged at 14,000 rpm for 20 min at 4 °C. Supernatants were carefully separated and transferred into new microcentrifuge tubes. Samples were dried by vacuum centrifugation and were stored at −80 °C until GC-MS analysis for flux measurements.

**Gel electrophoresis and immunoblotting**. At each time point during a time course, samples were removed from the incubator and 75 µl of RBCs (and medium) from each sample tube lysed in 250 µl of 2× LDS buffer (Life Technologies) and placed in a thermomixer heating block (Grant Instruments, model PHMT) for 10 min at 70 °C at 800 rpm. Immediately, samples were flash frozen in liquid nitrogen and stored at −80 °C until analysis. Samples were allowed to reach room temperature before analysis, which was performed as described previously[11,15,28]. Briefly, NuPAGE Novex 4–12% Bis-Tris gradient gels (Life Technologies) were loaded and run with a non-reducing MES SDS running buffer as per the manufacturer's guidelines. Protein transfer to nitrocellulose membrane was performed using the iBlot system (Life Technologies) with a standard P3, 7 min protocol. The nitrocellulose membrane was blocked for 1 h at room temperature in blocking buffer, composed of 0.5% w/w BSA, (Sigma A7906)/non-fat dried milk (Marvel) in Tris buffered saline/0.05% Tween-20 (TBST). After blocking, membranes were incubated with 1:10,000 anti-PRDX-SO$_{2/3}$ antiserum (Abcam ab16830), diluted in blocking buffer, overnight at 4 °C. The following day, blots were washed for 3 × 5 min in TBST and then incubated for 1 h at room temperature with a 1:10,000 HRP-conjugated secondary antiserum (Sigma A6154). Blots were washed 4 × 10 min in TBST before performing chemiluminescence detection using Immobilon Forte reagent (Millipore). Equal protein loading in each lane was checked with the aid of gels stained with Coomassie SimplyBlue SafeStain (Life Technologies). Coomassie stained gel images were obtained using an Odyssey system (Licor Biosciences). Immunoblot membranes were scanned using an Amersham Imager 600 (GE Healthcare). Quantification of images was performed using NIH Image J software[11,16].

**NADH and NADPH assays**. At each time point, samples were collected from the incubator and centrifuged for 2 min at 375 × g at 4 °C. RBC pellets were washed with ice-cold PBS twice and then centrifuged at 375 × g at 4 °C. Supernatant was discarded and the RBC pellet flash frozen in liquid nitrogen and stored at −80 °C. Extraction of metabolites was performed with the buffers supplied with commercial assay kits following the manufacturer's instructions (Abcam ab65348 for NADH and ab65349 for NADPH measurements). Colorimetric measurements were made at 450 nm absorbance and at 25 °C using a PerkinElmer Ensight multimode plate reader. NADH and NADPH concentrations were determined from a standard calibration curve according to the manufacturer's instructions. NADH/NADPH absorbance measurements were acquired using Kaleido pate reader software.

**Cell viability assays for metabolic inhibitor experiments**. RBCs were isolated from donors as mentioned above. The RBC pellet was made up to 9 ml with Krebs-Henseleit Buffer. 420 µl of RBCs were aliquoted into the wells of a 96 deep well plate for drug addition. 2.1 µl of diluted drug (1:200 dilution; 0.5% DMSO final) was added to each well and mixed thoroughly. For 6AN, concentrations were screened from 0 to 10 mM. For heptelidic acid, concentrations were screened over the range of 0 to 1 mM. Fifty microliter of RBCs preparation with compounds added (and control RBCs with only 0.5% DMSO added) were aliquoted into 96-well plates. The plates were incubated at 37 °C in constant darkness. Sampling was then performed every 24 h over a period of 96 h. At each time point, RBCs in a plate were re-suspended by pipetting up and down. The 96-well plates were then spun for 5 min at 375 × g and 25 µl of the supernatant transferred into a 384-well plate for assaying. A standard curve was prepared by lysing untreated cells with water (hypotonic lysis) and then performing serial dilutions in a 96-well plate. The absorbance of the sample supernatants was measured at 480 nm (in a 384-well plate) and using a Tecan M1000 plate reader. Red cell absorbance measurements were acquired using Tecan M1000 plate reader software.

**Red cell microscopy**. During the circadian time course with metabolic inhibitors, 10 µl of sample was collected at specific time points and spread onto a microscope slide (Thermo Scientific Superfrost Plus) and air dried. Microscopic images were captured using a Leica DM IL LED inverted microscope with LAS software version 4.8.

## Quantification and statistical analysis

**Mass spectrometry data processing, identification and metabolite enrichment analysis**. Vendor specific LC-MS raw data files from the mass spectrometer were extracted using Progenesis QI for metabolomics using parameters: feature detection = high resolution, peak processing = centroid data with 70,000 (FWHM)

resolution. In positive ionization mode, M + H, M + 2H, M + Na, M + NH$_4$, in negative ionization mode, M-H, M-2H, M + Na-2H were considered. Agilent Mass Hunter software v B.07.00 was used to extract GC-MS data. Features having a coefficient of variation (CV) lower than 30% among quality control samples were selected for downstream analyses and features having CV more than 30% were dropped from the dataset. We detected 1,698 features that had a coefficient of variation <30% in quality control samples. This included 533 features from negative mode LC-MS and 1074 from positive mode, with an additional 91 from GC-MS. Metabolite Set Enrichment Analysis was performed using Metaboanalyst software v 4.0, implemented in the R programming language.

Rhythmic metabolites were identified using retention time and MS/MS spectra of metabolite standards for LC-MS samples. Retention time and MS spectra from GC-MS analyzed samples were compared with metabolite standards and the National Institute of Standards and Technology (NIST) mass spectral library to confirm identification. The maximum number of metabolites detected in the RBC metabolome until now is 213[57]. The percentage of rhythmic metabolites in this study is 21% (46 rhythmic metabolites from total known RBC metabolome, i.e., 213 metabolites).

**Metabolic flux calculations**. Metabolic fluxes were estimated by using previously well-established models[23,58,61,62]. Briefly, the Pentose Cycle (PC) can be estimated using NMR by measuring differential enrichment of C3-lactate and C2-lactate after feeding cells with 2-$^{13}$C$_1$-Glucose with the formula:

$$\frac{C3 - \text{Lactate}}{C2 - \text{Lactate}} = \frac{2PC}{1 + 2PC} \qquad (1)$$

Where *PC* refers to fraction of glucose used to produce pentose phosphate pathway-derived Glyceraldehyde-3-phosphate.

Differential enrichment of $^{13}$C$_1$-lactate (m + 1) and $^{13}$C$_2$-lactate (m + 2) determined using GC-MS can be used to estimate fluxes after feeding cells with 1,2-$^{13}$C$_2$-Glucose with the formula:

$$\frac{(M + 1)\,\text{Lactate}}{(M + 2)\,\text{Lactate}} = \frac{3PC}{1 - PC} \qquad (2)$$

Metabolic fluxes for various metabolic pathways can be calculated from PC and glucose uptake as follows:

$$\text{PPP flux} = PC \times \text{Glucose consumption} \qquad (3)$$

$$\text{Glycolytic flux} = (1 - PC) \times \text{Glucose consumption} \qquad (4)$$

**Rhythmicity analysis**. Analysis of circadian waveforms was performed using two independent statistical methods[22], the Rhythmicity Analysis Incorporating Nonparametric method (RAIN)[18] and a harmonic regression method (ARSER)[22,63], which are implemented in the R programming language. P-value outputs are shown for each plot, along with the best-fitting period (e.g., for short or long period oscillations that deviate from 24 h). Only P-values for RAIN analyses are shown in the text and figures, but the alternative rhythm detection algorithm ARSER yielded results matching RAIN. Plots were produced in either R or GraphPad Prism (version 7 and 8). Statistical parameters such as details of replication and error bar meaning were reported in the figure legends.

**Bioluminescence assay analysis**. Exported images were quantified in a time series using NIH Image J software. Circadian rhythmicity in bioluminescence data was measured using a modified version of the R script "CellulaRhythm"[28,64].

**Reporting summary**. Further information on research design is available in the Nature Research Reporting Summary linked to this article.

## Data availability

All data are available from the authors upon request. Source Data are provided with the online version of the paper: for Fig. 3D it is provided in Supplementary Fig. 2 and Supplementary Fig. 3, that for Fig. 3G in Supplementary Fig. 4 and Supplementary Fig. 5, that for Fig. 5D in Figs. 5 and 6 in Supplementary Fig. 8, and that for Fig. 7C, D in Supplementary Fig. 9. Metabolomics data for Fig. 1 can be accessed on the Metabolights platform; Study submission code: MTBLS1285 and link https://www.ebi.ac.uk/metabolights/MTBLS1285. Source Data for flux experiments for Figs. 3B, C, E, F, 4A, B, 6D, E, 7B, D–F are provided in an Excel spreadsheet. The remaining data are available from the authors upon request. Source data are provided with this paper.

## Code availability

RAIN and ARSER codes adapted from Thaben and Westermark[18] and Yang et al.[63], respectively. Bioluminescence Circadian-R script "CellulaRhythm" was adapted from Rey et al.[28] and Hirota et al.[64].

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

## Acknowledgements

A.B.R. acknowledges funding from the Perelman School of Medicine, University of Pennsylvania, and the Institute for Translational Medicine and Therapeutics (ITMAT), Perelman School of Medicine, University of Pennsylvania. Research reported in this publication was supported by the National Institute of Diabetes and Digestive and Kidney Diseases (NIDDK) of the National Institutes of Health (NIH) under award number DP1DK126167 to A.B.R. The content is solely the responsibility of the authors and does not necessarily represent the official views of the National Institutes of Health. We thank J. Millar of the University of Pennsylvania Institute of Diabetes, Obesity and Metabolism (IDOM) Metabolic Tracing Core for carbon labeling experiments. A.B.R. also acknowledges funding from the European Research Council (ERC Starting Grant No. 281348, MetaCLOCK), the EMBO Young Investigators Programme, and the Lister Institute of Preventive Medicine. A.B.R. was supported in part by a Wellcome Trust Senior Fellowship in Clinical Science (100333/Z/12/Z) at the University of Cambridge, and also in part by the Francis Crick Institute, which receives its core funding from Cancer Research UK (FC001534), the UK Medical Research Council (FC001534), and the Wellcome Trust (FC001534). G.R. was supported by an Advanced SNSF Postdoctoral Mobility Fellowship and an EMBO Long-Term Fellowship. We thank P. Grice for assistance with NMR experiments, and J. Jones for help with blood sampling for preliminary experiments. We are also grateful for access to the MRC Biomedical NMR Centre at the Francis Crick Institute.

## Author contributions

R.C, G.R., and A.B.R. designed and planned the experiments. R.C., G.R., R.L., and S.R. performed cell experiments. P.K.J. performed mouse experiments with R.C. U.K.V. performed elements of the circadian data analysis. R.C. and S.R. performed the metabolite extractions. R.C., P.D., M.S.D.S., and D.M.M. performed NMR and mass spectrometry analyses for metabolomics samples, with oversight from J.I.M., A.M.W., and A.B.R. R.C. and A.B.R. wrote the manuscript, with contributions from all of the authors.

## Competing interests

The authors declare no competing interests.
