## [Peer Review File · Nature Communications]

Reviewer #1 (Remarks to the Author):

The authors show that human RBCs display circadian regulation of glucose metabolism required to sustain daily redox oscillations. They found daily rhythms of metabolite levels and flux through glycolysis and PPP and show that inhibition of critical enzymes in either pathway abolished rhythms. Furthermore, metabolic flux rhythms also occur in nucleated cells, and persist when the core transcriptional circadian clockwork is absent in Bmal1 knockouts. The amount of work presented here is large and the results are convincing and novel.

Oren Froy

Reviewer #2 (Remarks to the Author):

This study by Ratnasekhar Ch et al. is novel and significant because it reports that glycolysis and pentose phosphate pathway are coupled to circadian clock of peroxiredoxin oxidation in the mammalian red blood cells.

The authors identified that glycolysis and pentose phosphate pathway exhibit anti-phase circadian oscillations with respect to one another. The metabolic flux analysis performed throughout the paper is powerful and demonstrates clear circadian rhythms of glycolysis and pentose phosphate pathway. Additionally, authors identified that metabolic flux and the circadian rhythm of peroxiredoxin oxidation are coupled independent of major circadian regulation via transcription-translation feedback loops. The findings are important and interesting, and the study is well done with appropriate controls. However, to improve the manuscript there are several major and minor points that need to be addressed:

Major points:

- The authors do not appear to have provided the raw and/or processed data of untargeted LC-MS and GC-MS metabolomics analysis as well as metabolic flux analysis with NMR and GC-MS. The authors should include the data either in the supplement or deposit in a public database so that readers might re-analyze the data if desired.
- The authors mention that RBCs do not synthesize nucleotides, however, purine metabolism is one of the pathways in the enrichment analysis of Fig. 1D-E. Could authors describe which metabolites from purine metabolism are included in the enrichment analysis and if RBCs do not synthesize purines, why those metabolites are present in the analysis?
- The authors state that G6P exhibit anti-phase profile with glucose, however, Fig. 2B shows a positive correlation between glucose and G6P. Could authors explain or correct the discrepancy?
- Sample size (n) on Fig. 4D should be indicated.
- On Fig. 4D could authors justify the use of relatively short final time point (100 min) rather than a later final time point (e.g. 12 h)?
- On the HA treatment of Fig. 5D, if the first two data points (24 and 28 h) are ignored, there seems to be a circadian pattern although with an overall lower PRDX baseline compared with control.

Therefore, I disagree with the conclusion that HA treatment has caused PRDX-SO₂/3 arrhythmicity. Authors should either show data that more strongly shows effect of HA treatment on PRDX-SO₂/3 arrhythmicity or adjust their conclusion for this section.

Minor points:

- Authors should add annotations of metabolite on the heatmap of Fig. 1B.
- It is better to keep the colors consistent for representation of the positive and negative values on Fig. 1B and Fig. 2B. On Fig. 1A red represents positive values but on Fig. 2B red represents negative values.

Reviewer: Alireza Delfarah

Reviewer #3 (Remarks to the Author):

This manuscript entitled 'Rhythmic glucose metabolism regulates the redox circadian clockwork in human red blood cells' seeks to identify the source of previously identified conserved redox (peroxiredoxin oxidation) rhythms in red blood cells. In the absence of transcription and translation in RBCs, the authors reasoned that the source is likely due to rhythmic metabolism. To test this, the authors used metabolomics profiling by LC/GC-MS as well as NMR and found circadian rhythms of glycolysis and the pentose phosphate pathway (PPP). These pathways appear to oscillate anti-phasic to one another and PRDX oxidation parallels PPP activity. The authors further suggest that disruption of glycolysis or PPP affects PRDX oxidation rhythms as well as rhythms of NADH and NADPH. Finally, they demonstrate that rhythms of glycolytic flux (as well as PDRX oxidation) persist in nucleated cells even when clock transcription factors are inhibited. Overall, the authors make a convincing argument that glycolytic and PPP metabolism parallel PDRX oscillations and are likely a driving force behind them. This study adds significant clues to understanding the mechanisms controlling the conserved 'redox' clock.

Major comments:

- 1) If their model is correct, there should also be a circadian rhythm of ROS generation and detoxification in RBCs. These rhythms are presumably the link between glycolytic/PPP rhythms, NADPH generation and PDRX oxidation (ie PPP flux will generate a rhythm of NADPH which will in turn reduce hyperoxidized PDRX). It would be helpful to know whether RBCs display rhythms of Hb-derived ROS.
- 2) In figure 6E, the authors use protease and proteasome inhibitors to block PDRX oxidation rhythms. This is a very crude approach which will lead to many non-specific effects on protein and likely cell viability after multiple days of treatment.

Minor comments:

- 1) What is the mechanism of glycolytic vs PPP switching? Is it at the level of G6PD? It would be

interesting to measure G6PD protein levels over the circadian timescale (although it's unlikely to be rhythmic since there is no new protein synthesis).

2) Figure 5D: The PDRX-SO₂/3 blot for the HA treated cells does not look representative of the graphs above. There appears to be a robust rhythm of PDRX oxidation.

3) For the synchronized RBC time course experiments it is unclear how subjective day and night are determined. For example, in Figure 1A why are time points 1-12 shaded light and 12-24 shaded dark?

Point-by-Point Response to the Reviewers

Manuscript NCOMMS-20-30556: “Rhythmic glucose metabolism regulates the redox circadian clockwork in human red blood cells”

Reviewer #1 (Remarks to the Author):

The authors show that human RBCs display circadian regulation of glucose metabolism required to sustain daily redox oscillations. They found daily rhythms of metabolite levels and flux through glycolysis and PPP and show that inhibition of critical enzymes in either pathway abolished rhythms. Furthermore, metabolic flux rhythms also occur in nucleated cells, and persist when the core transcriptional circadian clockwork is absent in Bmal1 knockouts. The amount of work presented here is large and the results are convincing and novel.

Thank you for your comments on the manuscript.

Reviewer #2 (Remarks to the Author):

This study by Ratnasekhar Ch et al. is novel and significant because it reports that glycolysis and pentose phosphate pathway are coupled to circadian clock of peroxiredoxin oxidation in the mammalian red blood cells.

The authors identified that glycolysis and pentose phosphate pathway exhibit anti-phase circadian oscillations with respect to one another. The metabolic flux analysis performed throughout the paper is powerful and demonstrates clear circadian rhythms of glycolysis and pentose phosphate pathway. Additionally, authors identified that metabolic flux and the circadian rhythm of peroxiredoxin oxidation are coupled independent of major circadian regulation via transcription-translation feedback loops. The findings are important and interesting, and the study is well done with appropriate controls. However, to improve the manuscript there are several major and minor points that need to be addressed:

Thank you for your useful comments and suggestions for improving the manuscript.

Major points:

- The authors do not appear to have provided the raw and/or processed data of untargeted LC-MS and GC-MS metabolomics analysis as well as metabolic flux analysis with NMR and GC-MS. The authors should include the data either in the supplement or deposit in a public database so that readers might re-analyze the data if desired.

Thank you for the suggestion. The untargeted metabolomics data have been submitted to the Metabolights platform with submission code: MTBLS1285. The flux data are now available in Supplementary Information.

- The authors mention that RBCs do not synthesize nucleotides, however, purine metabolism is one of the pathways in the enrichment analysis of Fig. 1D-E. Could authors describe which metabolites from purine metabolism are included in the enrichment analysis and if RBCs do not synthesize purines, why those metabolites are present in the analysis?

Although RBCs do not to synthesize purine nucleotides de novo – they are produced in salvage reactions (Dudzinska et al., 2006). In this study, we found the synthesis of inosine and inosine monophosphate from uniform labelling experiments. We found circadian rhythms of only two metabolites, uridine and UDP.

- The authors state that G6P exhibit anti-phase profile with glucose, however, Fig. 2B shows a positive correlation between glucose and G6P. Could authors explain or correct the discrepancy?

Thank you for the suggestion. We have corrected this discrepancy in the text accordingly:

“Glucose displayed a well-defined 24-hour oscillation ($P = 0.005$ by RAIN), with glucose-6-phosphate (G6P) showing a similar phase, albeit with reduced rhythmicity ($P = 0.0369$ by RAIN) (Fig. 2A). Importantly, ribose-5-phosphate (R5P), a principal metabolite in the PPP, exhibited daily cycling in anti-phase to G6P (Fig. 2A-B).”

- Sample size (n) on Fig. 4D should be indicated.

We state the sample size ($n = 3$ biological replicates) in the Fig. 4D legend.

- On Fig. 4D could authors justify the use of relatively short final time point (100 min) rather than a later final time point (e.g. 12 h)?

The goal of this experiment was to measure metabolic fluxes at opposite phases of the circadian clock in live mice. Therefore, we collected blood from mice at opposite time points in the daily cycle, and then subjected the purified RBCs to metabolic flux measurements. The reason to chose a short time frame was to measure metabolic fluxes as close as possible to the blood collection time.

- On the HA treatment of Fig. 5D, if the first two data points (24 and 28 h) are ignored, there seems to be a circadian pattern although with an overall lower PRDX baseline compared with control. Therefore, I disagree with the conclusion that HA treatment has caused PRDX-SO₂/3 arrhythmicity. Authors should either show data that more strongly shows effect of HA treatment on PRDX-SO₂/3 arrhythmicity or adjust their conclusion for this section.

Thank you for this comment. The HA-treated PRDX profile peaks around CT40, CT52-CT56 and CT72. It is therefore difficult to argue for a 24h rhythm in this profile, especially given the statistical p-value of 0.231. However, there may be ultradian rhythmicity appearing instead (12h to 18h oscillations).

Minor points:

- Authors should add annotations of metabolite on the heatmap of Fig. 1B.

Annotation of metabolites are presented in Fig. 1B as per the reviewer’s suggestion.

- It is better to keep the colors consistent for representation of the positive and negative values on Fig. 1B and Fig. 2B. On Fig. 1A red represents positive values but on Fig. 2B red represents negative values.

Thank you for this suggestion. We have changed to colours for consistency between these figures.

Reviewer #3 (Remarks to the Author):

This manuscript entitled ‘Rhythmic glucose metabolism regulates the redox circadian clockwork in human red blood cells’ seeks to identify the source of previously identified conserved redox (peroxiredoxin oxidation) rhythms in red blood cells. In the absence of transcription and translation in RBCs, the authors reasoned that the source is likely due to rhythmic metabolism. To test this, the authors used metabolomics profiling by LC/GC-MS as well as NMR and found circadian rhythms of glycolysis and the pentose phosphate pathway (PPP). These pathways appear to oscillate anti-phasic to one another and PRDX oxidation parallels PPP activity. The authors further suggest that disruption of glycolysis or PPP affects PRDX oxidation rhythms as well as rhythms of NADH and NADPH. Finally, they demonstrate that rhythms of glycolytic flux (as well as PDRX oxidation) persist in nucleated cells even when clock transcription factors are inhibited. Overall, the authors make a convincing argument that glycolytic and PPP metabolism parallel PDRX oscillations and are likely a driving force behind them. This study adds significant clues to understanding the mechanisms controlling the conserved ‘redox’ clock.

We thank the reviewer for their useful comments and suggestions.

Major comments:

1) If their model is correct, there should also be a circadian rhythm of ROS generation and detoxification in RBCs. These rhythms are presumably the link between glycolytic/PPP rhythms, NADPH generation and PDRX oxidation (ie PPP flux will generate a rhythm of NADPH which will in turn reduce hyperoxidized PDRX). It would be helpful to know whether RBCs display rhythms of Hb-derived ROS.

We thank reviewer for the suggestion. Due to high risk associated with COVID-19 infection and handling blood samples at the present time, it would be extremely difficult to perform this experiment. However, below we present data that substantiates the assertion that RBCs display ROS rhythms.

*Peroxiredoxins are major antioxidant enzymes that control intracellular ROS (Hall et al., 2009), and in the process of reducing ROS, the catalytic cysteine of 2-Cys PRDXs is oxidised to sulfenic acid (PRDX-SO), which in turn is over/hyper-oxidized to sulfinic acid form (PRDX-SO_{2/3}). This renders PRDX catalytically inactive during the elimination of H₂O₂, a critical intracellular ROS. Reversible inactivation of H₂O₂ by PRDX2 occurs in RBCs (Cho et al., 2014). This indicates that ROS and peroxiredoxin oxidation are tightly coupled and dependent on each other. Previous results show circadian oscillations in peroxiredoxin and in autoxidation of haemoglobin (fig 1), which is the key source generating ROS (O’Neill and Reddy, 2011). Coupled to this, increasing the concentration of H₂O₂ in cultured RBCs leads to increased oxidation of peroxiredoxin (fig 2) (O’Neill and Reddy, 2011)). Recent data support the notion of endogenous circadian H₂O₂ (i.e. ROS) oscillation in mammalian cells (U2OS cells) (fig 3) (Pei et al., 2019). The same is true in plants (*Arabidopsis thaliana*; fig 4) (Lai et al., 2012). In all of these systems, rhythmic peroxiredoxin oxidation is conserved (Edgar et al., 2012).*

Fig 1: Circadian rhythms in haemoglobin oxidation (figure obtained from Nature 2011, 469(7331):498-503; PMID 21270888)

Fig 2: RBCs exposed to H₂O₂ (0-1000 μM H₂O₂) in vitro (figure obtained from Nature 2011 Jan 27;469(7331):498-503; PMID 21270888)

Fig 3: Circadian rhythms in H₂O₂ levels in Human U2OS cells. (Endogenous H₂O₂ levels in living cells are measured using a genetically encoded physiological H₂O₂ sensor HyPerRed) (figure obtained from Nat Cell Biol 2019, 21(12):1553-1564; PMID: 31768048)

Fig 4: Circadian rhythms in H₂O₂ levels in plant *Arabidopsis thaliana* (figure obtained from Proc Natl Acad Sci U S A. 2012 Oct 16;109(42):17129-34; PMID: 23027948).

2) In figure 6E, the authors use protease and proteasome inhibitors to block PDRX oxidation rhythms. This is a very crude approach which will lead to many non-specific effects on protein and likely cell viability after multiple days of treatment.

We understand that the use of chemical inhibitors has inherent limitations. Therefore, we chose to use two orthogonal approaches, proteasome inhibitors and PRDX inhibitors, to block PRDX oxidation rhythms. The first approach uses a PRDX oxidation inhibitor, conoidin A, that prevents overoxidation of PRDX and has been shown to abolish PRDX oxidation rhythms. The second approach uses the same proteasome inhibitor (MG-132), which Cho et al. (2014) showed is sufficient to abolish PRDX oxidation rhythms in RBCs (Cho et al., 2014). The fact that both approaches lead to a similar effect on metabolic fluxes gives us confidence that these are not non-specific effects.

Minor comments:

1) What is the mechanism of glycolytic vs PPP switching? Is it at the level of G6PD? It would be interesting to measure G6PD protein levels over the circadian timescale (although it's unlikely to be rhythmic since there is no new protein synthesis).

Thank you for this interesting question. In a separate study, we performed circadian proteomic analyses in RBCs (unpublished) and did not find obvious rhythmicity in metabolic proteins, which could potentially happen due to selective degradation over time. It is therefore likely that the switching mechanism is at the post-translational level. Unfortunately, as far as we know, there are no good reagents yet available to test possibly relevant post-translational modifications of G6PD.

2) Figure 5D: The PDRX-SO2/3 blot for the HA treated cells does not look representative of the graphs above. There appears to be a robust rhythm of PDRX oxidation.

*Thank you for this comment. The HA-treated PRDX profile peaks around CT40, CT52-CT56 and CT72. It is therefore difficult to argue for a 24h rhythm in this profile, especially given the statistical *p*-value of 0.231. However, there may be ultradian rhythmicity appearing instead (12h to 18h oscillations).*

3) For the synchronized RBC time course experiments it is unclear how subjective day and night are determined. For example, in Figure 1A why are time points 1-12 shaded light and 12-24 shaded dark?

For the synchronization of RBCs, we used a temperature cycle of 37°C (for 12 hours) and 32°C (for 12 hours) for the initial 24 hours, and then left them in constant temperature of 37°C. Since human core body temperature rises in the day time (equivalent to 37°C phase), we arbitrarily showed subsequent subjective day and night as light grey and dark grey relative to these respective initial phases.

References:

Cho, C.-S., Yoon, H.J., Kim, J.Y., Woo, H.A., and Rhee, S.G. (2014). Circadian rhythm of hyperoxidized peroxiredoxin II is determined by hemoglobin autoxidation and the 20S proteasome in red blood cells. *Proc National Acad Sci* *111*, 12043–12048.

Dudzinska, W., Hlynczak, A.J., Skotnicka, E., and Suska, M. (2006). The purine metabolism of human erythrocytes. *Biochem Mosc* *71*, 467–475.

Edgar, R.S., Green, E.W., Zhao, Y., Ooijen, G. van, Olmedo, M., Qin, X., Xu, Y., Pan, M., Valekunja, U.K., Feeney, K.A., et al. (2012). Peroxiredoxins are conserved markers of circadian rhythms. *Nature* 485, 459–464.

Hall, A., Karplus, P.A., and Poole, L.B. (2009). Typical 2-Cys peroxiredoxins – structures, mechanisms and functions. *Febs J* 276, 2469–2477.

Lai, A.G., Doherty, C.J., Mueller-Roeber, B., Kay, S.A., Schippers, J.H.M., and Dijkwel, P.P. (2012). CIRCADIAN CLOCK-ASSOCIATED 1 regulates ROS homeostasis and oxidative stress responses. *Proc National Acad Sci* 109, 17129–17134.

O'Neill, J.S., and Reddy, A.B. (2011). Circadian clocks in human red blood cells. *Nature* 469, 498–503.

Pei, J.-F., Li, X.-K., Li, W.-Q., Gao, Q., Zhang, Y., Wang, X.-M., Fu, J.-Q., Cui, S.-S., Qu, J.-H., Zhao, X., et al. (2019). Diurnal oscillations of endogenous H₂O₂ sustained by p66Shc regulate circadian clocks. *Nat Cell Biol* 21, 1553–1564.

Reviewer #2 (Remarks to the Author):

The authors have addressed all of my concerns.

Reviewer #3 (Remarks to the Author):

The authors have adequately responded to my comments. I recommend this manuscript for publication.